

# Disparate evolution mechanisms and optical absorption for transboundary soot particles passing through inland and sea pathways

Jian Zhang[1], Zexuan Zhang[2], Keliang Li[2], Xiyao Chen[2], Xinpeng Xu[2], Yangmei Zhang[3], Anzhou Han[4], Yuanyuan Wang[5], Jing Ding[1], Liang Xu[6], Yinxiao Zhang[7], Hongya Niu[8], Shoujuan Shu[2], and Weijun Li[2,*]

[1]School of Environmental and Material Engineering, Yantai University, Yantai 264005, China

[2]Key Laboratory of Geoscience Big Data and Deep Resource of Zhejiang Province, Department of Atmospheric Sciences, School of Earth Sciences, Zhejiang University, Hangzhou 310027, China

[3]Key Laboratory of Atmospheric Chemistry, China Meteorological Administration, Beijing 100081, China

[4]Trier College of Sustainable Technology, Yantai University, Yantai 264005, China

[5]Hangzhou International Innovation Institute of Beihang University, Hangzhou 310000, China

[6]College of Sciences, China Jiliang University, Hangzhou 310018, China

[7]Flight College, Shandong University of Aeronautics, Binzhou 256600, China

[8]Key Laboratory of Resource Exploration Research of Hebei Province, Hebei University of Engineering, Handan 056038, China

[*]Corresponding Email: liweijun@zju.edu.cn (W. J. Li)





## Abstract

Soot particles, as a type of warming aerosols, play a critical role in climate warming. During transport, these particles undergo atmospheric condition-dependent aging processes that influence their microphysical and optical properties. Here, we investigated the variations in morphology, mixing states, sizes, and optical absorption of soot-containing particles and further revealed their evolution mechanisms during two distinct transboundary transport through the inland and sea pathways. Comparing transboundary soot-containing particles transported through the inland and sea pathways, we found more soot cores in the latter individual particles, although their dominant mixing states exhibited a similar transition from partly-coated at 62-67% by number to embedded structures at 71-72%. The core-shell size ratio ($D_p/D_c$) and soot core fractal dimension of embedded soot-containing particles transported through the sea pathway were both greater compared to the inland pathway. These differences were attributed to distinct evolution mechanisms experienced by soot-containing particles during transport: heterogeneous aging processes through the inland pathway and cloud processes through the sea pathway. Optical simulation showed amplified light absorption of soot-containing particles during their transboundary transport. Furthermore, the radiative absorption amplification per unit $D_p/D_c$ change reduced by 72% due to the entrainment of multiple soot cores within individual particles following the transport pathway change from the inland to the sea. This study suggests varied mixing configurations and radiative absorption of transboundary soot-containing particles driven by different environmental conditions and highlights the necessity of incorporating multicore black carbon mixing structures into climate models.





## 1. Introduction

In recent years, eastern China still faces severe haze pollution in winter despite a series of control policies adopted by the local government (Peng et al., 2021; Zang et al., 2022; Zhai et al., 2021). High concentrations of fine haze particles not only affect human health (Geng et al., 2021; Zhang et al., 2017b), but also influence the global climate (Li et al., 2016a; Suzuki and Takemura, 2019). In addition to high emissions of pollutants and stable meteorological conditions (Niu et al., 2016; Zhao et al., 2019; Zheng et al., 2015), transboundary transport of pollutants has been confirmed to be an important factor causing the formation of regional haze pollution in eastern China in winter (Li et al., 2019; Zhang et al., 2021; Zhang et al., 2019a). For example, large amounts of haze pollutants in the North China Plain (NCP) can be transported thousands of kilometers to the Yangtze River Delta (YRD) under cold fronts to induce long-lasting regional haze events (Huang et al., 2020).

In eastern China, the NCP and the YRD as two rapid economic developed regions suffer the heaviest haze pollution (Zhong et al., 2019). Recent studies found frequent transboundary transport of pollutants from the NCP to the YRD using various methods including field observation and model simulation (Kang et al., 2019; Li et al., 2019; Xie et al., 2023; Yan et al., 2024; Zhang et al., 2021). For instance, Kang et al. (2019) utilized numerical models to show that the $PM_{2.5}$ contribution from the NCP reached to ~30% in the YRD under cold fronts. Field results showed that concentrations of organic matter and secondary inorganic ions in $PM_{2.5}$ in the YRD increased by approximately 1-2 times following the invasion of haze pollutants from the NCP (Zhang et al., 2021). Xie et al. (2023) also suggested that carbonaceous aerosols and secondary inorganic ions were effectively transported from the NCP to the YRD based on the simulation of the average atmospheric age of haze particles. During the transboundary transport, abundant secondary aerosols (e.g., sulfate and nitrate) can be formed through heterogeneous chemical processes, influencing the particulate hygroscopicity (Li et al., 2019; Li et al., 2025; Zhang et al., 2021). Although these studies revealed changes in the bulk chemical composition of fine particles and main formation processes of



secondary aerosols during the transport, there is a lack of evolution mechanisms of
transboundary particles and their potential health or climate effects.

We noticed that cold fronts had different pathways to convey air pollutants from

the NCP to the YRD depending on the location of high-pressure systems. The previous
studies showed that cold fronts from the high-pressure system locating in the west of
the NCP normally transported haze pollutants across city clusters in eastern China (Hou
et al., 2020; Jin et al., 2021). If the high-pressure system located in the interior of the
NCP, heavy haze covering the Jing-Jin-Ji region (i.e., Beijing, Tianjin, and Hebei) could
move out from inland China to the East China Sea and return into the inland region
under prevailing winds, influencing air quality of the YRD (see section 3.1). These haze
movements from the NCP to the YRD were clearly observed from the satellites
(https://worldview.earthdata.nasa.gov). To our knowledge, the previous studies only
focused on the haze transportation from the NCP to the YRD through the inland
pathway based on field measurements and model simulation (Huang et al., 2020; Yan
et al., 2024). Obviously, there was a bench of data available from national ground
monitoring net station of air quality to support the measurements and modelling studies.
However, transboundary haze pollutants crossed the East China Sea remain unexplored.
We expect different chemical mechanisms and aging processes in gas-aerosol
interactions in two haze layers because of different meteorological factors (e.g., relative
humidity) and pollutant emissions in transport pathways.

Soot particles (i.e., black carbon, also called elemental carbon) emitted from

incomplete burning of fossil fuels and biomass are important light absorbing aerosols
in fine particles, exerting favorable effects on global warming in the atmosphere (Bond
et al., 2013; Cappa et al., 2012; Jacobson, 2001). Soot particles serve as an excellent
tracer to reflect atmospheric aging because their morphology (Fierce et al., 2020; Wu
et al., 2018; Yuan et al., 2019), mixing states (Wang et al., 2019; Wang et al., 2016),
sizes (Adachi et al., 2014; Xu et al., 2020), and mass (Liu et al., 2020; Zhang et al.,
2018) can be significantly altered during transport. Previous global studies examining
pollutant transport, such as trans-oceanic dust events affecting East Asia (Xu et al., 2020)

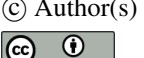



and North America outflow influencing the Azores in the North Atlantic (China et al.,
2015), have extensively utilized soot particles as a primary investigative target to
understand environmental impacts. The transport corridor from the NCP to the YRD
represents no exception, where soot particles persist as a critical, abundant component
of atmospheric aerosols (Huang et al., 2020; Zhang et al., 2023). Compared to transport
over inland China, soot particles transported to the YRD from the NCP via the East
China Sea may encounter more humid conditions. These distinct atmospheric
environments can lead to different alterations in soot physicochemical properties (Li et
al., 2024). Therefore, the evolution of soot particles and their environmental and climate
impacts should be examined in different synoptic weather processes. Although
simulating soot climate effect is readily achievable in models, these simulations often
overlook alterations in soot microphysical properties due to limited studies, thereby
introducing considerable uncertainties into the results (IPCC, 2021).

To bridge this critical gap, we observed two types of transboundary transport

events of haze pollutants (passing through the inland and through the sea pathways)
from the NCP to the YRD and first compared physicochemical characteristics (e.g.,
morphology, mixing states, and sizes) of transboundary soot particles in the two events
by various microscopic measurements. Based on microscopic observations, the
evolution mechanisms of two types of transboundary soot particles were unveiled.
Using a novel optical calculation model, we further estimated the change in soot optical
absorption between two types of transboundary transport. This study aims to emphasize
how divergent transport environments (inland vs. sea) impart distinct physicochemical
effects on soot particles. This promotes understanding of the weather-aerosol-pollution-
climate nexus, ultimately informing more accurate prediction of soot role in regional
climate forcing and atmospheric chemistry.

## 2. Methods

### 2.1 Sample collection

Three sites (Beijing, Handan, and Zhengzhou) in the NCP and two sites (Nanjing

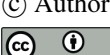



and Hangzhou) in the YRD were selected to collect ambient $PM_{2.5}$ and individual
aerosol particles in December 2017 and 2020 based on the transport behavior of
airborne pollutants under cold fronts in winter (Figure S1). Beijing, Handan, and
Zhengzhou are located in the northern, central, and southern parts of the NCP,
respectively. Beijing is a typical megacity and influenced by local emissions and
regional transport, while Handan and Zhengzhou are two typical industrial cities and
affected by local industrial, vehicular, and residential emissions. Nanjing and Hangzhou
are two megacities located in the northern and southern parts of the YRD. These two
megacities can be influenced when haze particles in the NCP invade the YRD.
Therefore, the above five cities in the NCP and YRD are representative for exploring
transboundary transport of aerosols (e.g., soot particles) in haze plumes. The detailed
information of the sampling sites has been described in previous studies (Zhang et al.,
2023; Zhang et al., 2021).

Ambient $PM_{2.5}$ and blank samples (no pumping) were collected on preheated

quartz filters with 90 mm diameters (600°C for 4 hr, Whatman) twice a day from 8:30
(local time) to 20:00 and from 20:30 to 8:00 the next day using medium volume
samplers (TH-16A, Wuhan Tianhong, 100 L min$^{-1}$). Individual aerosol particles were
sampled on transmission electron microscopy (TEM) grids and silicon wafers four
times a day at around 2:00 (local time), 9:00, 14:00, and 20:00 utilizing individual
particle samplers (DKL-2, Genstar, 1 L min$^{-1}$) equipped with a 0.5 mm jet nozzle
impactor. To avoid particles overlapping on the substrate, the sampling duration of
individual particles needs to be adjusted from 30 s to 15 min according to current $PM_{2.5}$
concentrations. Hourly $PM_{2.5}$ concentrations and meteorological parameters including
relative humidity (RH) and winds were derived from local monitoring stations
(https://www.aqistudy.cn/).

**2.2 $PM_{2.5}$ and individual particle analyses**

Water-soluble inorganic ions (i.e., $NO_3^-$, $SO_4^{2-}$, $NH_4^+$, $F^-$, $Cl^-$, $Na^+$, $K^+$, $Ca^{2+}$, and

$Mg^{2+}$), carbonaceous components (organic and elemental carbon), and trace metallic



elements in PM$_{2.5}$ samples were analyzed using an ion chromatography system (Dionex
ICs-90, USA), an OC/EC analyzer (Sunset Laboratory, USA), and inductively coupled
plasma mass spectrometry (ICP-MS, Agilent 7500ce). The detailed experimental
processes can be found in previous studies (Pan et al., 2013; Zhang et al., 2017a). In
this study, organic matter (OM) concentrations were obtained by multiplying organic
carbon (OC) concentrations by 1.91 reported by Xing et al. (2013).

Morphology, mixing states, and compositions of individual aerosol particles

collected on TEM grids were examined by TEM equipped with energy-dispersive X-
ray spectrometry (EDS) (JEM-2100, JEOL). The acquisition time of TEM images and
EDS spectra is usually controlled within 15 s because of the damage of electron beams
to non-refractory aerosols. To better observe soot mixing states and measure soot
geometrical parameters, we enhanced the electron beam to sublime non-refractory
coatings of indiscernible soot cores after conventional TEM observations. Copper
element was excluded from particle EDS spectra because TEM grids are made of
copper (Li et al., 2025). Using an image analysis software (Radius, EMSIS GmbH), we
further obtained the equivalent circle diameter (ECD), length, and area of particles in
TEM images. In this work, 3642 individual particles were analyzed by TEM in total.

Three-dimensional morphology of individual particles collected on silicon wafers

was probed by atomic force microscopy (AFM, Dimension Icon) in tapping mode.
Employing a professional image analysis software (NanoScope Analysis), the bearing
area (A) and the bearing volume (V) of particles in AFM images were quantified. The
ECD and the equivalent sphere diameter (ESD) of these particles can be calculated
applying equations (1) and (2).

$$ECD = \sqrt{\frac{4A}{\pi}} \tag{1}$$

$$ESD = \sqrt[3]{\frac{6V}{\pi}} \tag{2}$$

Figure S2 shows that there is a good correlation between the ESD and the ECD

with slopes at 0.62 for passing through the inland pathway transport and 0.39 for





passing through the sea pathway transport. According to the relationship between the
ESD and the ECD, ESDs of particles observed by TEM were computed. The ESDs of
soot particles were perceived as equivalent to their ECDs because they are composed
of solid carbonaceous spheres that are not affected by substrates in terms of morphology
(Barone et al., 2012; Li et al., 2016b).
The size ratio of soot-containing particles to their soot cores ($D_p/D_c$) was evaluated
using equation (3):

$$D_p/D_c = \frac{ESD_{soot\text{-}containing}}{ESD_{soot}} \tag{3}$$

where $ESD_{soot\text{-}containing}$ is the ESD of soot cores with their coatings and $ESD_{soot}$ is the
soot core ESD.
Based on the scaling law in following equations, we obtained the fractal dimension
($D_f$) of soot particles, which can be used to reflect the compactness of soot particles:

$$N = k_g \left(\frac{2R_g}{d_p}\right)^{D_f} \tag{4}$$

$$N = k_a \left(\frac{A_a}{A_p}\right)^{\alpha} \tag{5}$$

$$\delta = \frac{2a}{l} \tag{6}$$

$$\frac{L_{max}}{2R_g} = 1.50 \pm 0.05 \tag{7}$$

where N is the monomer number in soot particles, $k_g$ is the fractal prefactor, $R_g$ is the
gyration radius of soot particles, $d_p$ is the average diameter of soot monomers, $A_a$ is the
projected area of soot particles, $A_p$ is the average projected area of soot monomers, $k_a$
is a constant, $\alpha$ is an empirical projected area exponent, $\delta$ is the overlap parameter of
soot monomers, a is the average radius of adjacent soot monomers, l is the spacing of
adjacent soot monomers, and $L_{max}$ is the maximum length of soot particles. $k_a$ and $\alpha$
depend on $\delta$ (Oh and Sorensen, 1997).

**2.3 Meteorological fields and geographic sources**

Wind fields coupled with surface PM$_{2.5}$ concentrations covering eastern China





were obtained from European Centre for Medium-Range Weather Forecasts (ECMWF,
https://earth.nullschool.net/). Meteorological fields including winds and relative
humidity (RH) in eastern China at 1000 hPa were simulated using 1° × 1° Final
Reanalysis Data (FNL) from the National Centers for Environmental Prediction (NCEP,
https://rda.ucar.edu/datasets/ds083.2/).

The potential geographic sources of PM$_{2.5}$ at observation sites (Nanjing and

Hangzhou) in the YRD were identified based on the concentration-weighted trajectory
(CWT) analysis. In this study, 72 hr of air mass backward trajectories simulated from
the wind data sets in the Nation Oceanic Atmospheric Administration (NOAA,
ftp://arlftp.arlhq.noaa.gov/pub/archives/gdas1) were used for the CWT analysis. The
resolution of CWT trajectories consists of thousands of grid cells is 0.3° × 0.3°. The
equation for the CWT analysis is as follows:

$$C_{ij} = \frac{1}{\sum_{k=1}^{N} \tau_{ijk}} \sum_{k=1}^{N} C_k \tau_{ijk} \qquad (8)$$

where $C_{ij}$ is the average PM$_{2.5}$ concentration in a grid cell (i, j); $C_k$ is the measured PM$_{2.5}$
concentration for the trajectory k arriving at the observation site; $\tau_{ijk}$ is the number of
trajectory endpoints in the grid cell (i, j) for the $C_k$ sample; N is the number of samples
with trajectory endpoints in the grid cell (i, j).

In the CWT analysis, a weighing function as shown in equation (9) was applied to

further improve the CWT accuracy:

$$W = \begin{cases} 1 & \text{for } \log(n+1) \geq 0.85 \times \max_{\log(n+1)} \\ 0.725 & \text{for } 0.6 \times \max_{\log(n+1)} \leq \log(n+1) < 0.85 \times \max_{\log(n+1)} \\ 0.475 & \text{for } 0.35 \times \max_{\log(n+1)} \leq \log(n+1) < 0.6 \times \max_{\log(n+1)} \\ 0.175 & \text{for } \log(n+1) < 0.35 \times \max_{\log(n+1)} \end{cases} \qquad (9)$$

where $\log(n+1)$ is the density of trajectories.

**2.4 Optical calculation**

The Electron-Microscope-to-BC-Simulation (EMBS) tool developed by Wang et

al. (2021) was used to model morphology and mixing states of soot particles. The
EMBS tool capable of building various soot-containing particle models can be applied



in DDSCAT 7.3 to calculate soot optical properties based on the discrete dipole
approximation (DDA). DDA is completely flexible to the geometry of object particles
under the condition that the inter dipole separation d follows $|m|kd < 0.5$ and $k = 2\pi/\lambda$,
where m is the refractive index of particles and $\lambda$ is the incident light wavelength. To
minimize DDA uncertainty, the dipole size is much smaller than the soot monomer size.
Based on sizes and mixing states of soot-containing particles as well as $D_f$ and numbers
of soot cores obtained from microscopic analyses, we employed EMBS and DDSCAT
7.3 to calculate the light absorption enhancement ($E_{abs}$) of soot-containing particles
relative to their soot cores at 550 nm $\lambda$. In this study, soot-containing particles with one,
two, and three or more soot cores were distinguished. The volumes of soot cores and
their coatings remained constant in the optical calculation. The refractive indices of soot
cores and coatings were considered as $1.85 + 0.71i$ (Bond and Bergstrom, 2006) and
$1.53 + 0i$ (Worringen et al., 2008), respectively. Details about the EMBS and DDSCAT
7.3 can be found in the previous study (Wang et al., 2021).

## 3. Results and discussion

### 3.1 Determination of two types of transport models

Figures S3a-b show variations in hourly winds and $PM_{2.5}$ concentrations at the
observation sites in the NCP and the YRD from December 28 to 31, 2017 and from
December 5 to 8, 2020. The prevailing wind significantly changed from weak southern
winds to strong northern winds in the NCP and the YRD on December 30, 2017 and
December 7, 2020 under cold fronts (Figures S3a-b). On the same day, the average
$PM_{2.5}$ concentration in the NCP rapidly decreased from 318 $\mu g\ m^{-3}$ during the first
observation period and 179 $\mu g\ m^{-3}$ during the second observation period to 33 $\mu g\ m^{-3}$
and 37 $\mu g\ m^{-3}$ (Figures S3a-b). After 6-9 hours, the average $PM_{2.5}$ concentration in the
YRD suddenly increased from 62 $\mu g\ m^{-3}$ during the first observation period and 51 $\mu g$
$m^{-3}$ during the second observation period to 308 $\mu g\ m^{-3}$ and 113 $\mu g\ m^{-3}$ (Figures S3a-
b). Similar changes in $PM_{2.5}$ concentrations accompanied by winds were also found in
many transboundary transport events of pollutants (Wu et al., 2022; Xie et al., 2023;



Yan et al., 2024; Zhao et al., 2021). As a result, we inferred that there was a typical
transboundary transport process of pollutants from the NCP to the YRD on December
30-31, 2017 and December 7-8, 2020, respectively.

Figure 1 displays meteorological fields coupled with surface $PM_{2.5}$ concentrations

in eastern China during two transboundary transport events of pollutants. In the first
transport event, the wind blew from the NCP through the inland pathway towards the
YRD under the high-pressure system located in the west of the NCP (Figures 1a, 1c,
and S3c). It is interesting that there was a significant change in the wind field following
the high-pressure system movement to the interior of the NCP during the second
transport event compared to the first event, manifested as the wind mainly blowing from
the NCP to the East China Sea and then to the YRD (Figures 1b, 1d, and S3d). Although
a previous study also discovered comparable wind fields between the NCP and the YRD
using a weather model, the changes in chemical compositions and microphysical
properties of haze particles have not been defined during the transboundary transport
(Wu et al., 2022). To determine whether the transport pathway of pollutants was
consistent with the wind field, the $PM_{2.5}$ transport pathway was simulated based on the
CWT analysis (Figure 2). Figure 2 shows that $PM_{2.5}$ in Nanjing and Hangzhou was
mainly transported from the NCP through the inland pathway during the first transport
event but through the sea pathway during the second transport event. Therefore, we
concluded two transport models of haze pollutants from the NCP to the YRD, namely
passing through the inland and through the sea pathways.

During two transboundary transport events, concentrations of chemical

compositions in $PM_{2.5}$ in the NCP and the YRD significantly changed (Figure S4). The
concentration of secondary inorganic ions in the NCP decreased from 92-126 μg m$^{-3}$
during the polluted period to 28-30 μg m$^{-3}$ during the clean period (Figure S4). OM and
EC concentrations in the NCP also decreased from 43-76 μg m$^{-3}$ and 1.5-2.1 μg m$^{-3}$
during the polluted period to 17-31 μg m$^{-3}$ and 0.7-0.9 μg m$^{-3}$ during the clean period
(Figure S4). Following the transportation of large amounts of pollutants from the NCP
to the YRD, the concentrations of secondary inorganic ions, OM, and EC in the YRD



increased from 28-37 µg m$^{-3}$, 13-19 µg m$^{-3}$, and 1.0-1.4 µg m$^{-3}$ during the clean period
to 63-65 µg m$^{-3}$, 32-36 µg m$^{-3}$, and 1.6-2.7 µg m$^{-3}$ during the polluted period,
respectively (Figure S4). These results suggest that massive primary and secondary
aerosols including EC (i.e., soot) were transported from the NCP to the YRD under cold
fronts, both through the inland and the sea pathways.
Based on simulated meteorological fields, we noticed that polluted air masses
passing through the sea pathway underwent wetter environment during transboundary
transport compared to that passing through the inland pathway (Figures 1c-d). Table S1
also shows much higher average RH at 90% in the YRD following the transport of haze
pollutants from the NCP to the YRD through the sea pathway in contrast to the inland
pathway (RH = 83%). High RH can contribute to the transformation of microphysical
properties (e.g., mixing states, sizes, and morphology) of soot particles in the
atmosphere, but the reaction mechanism may vary under different high RH levels (Fu
et al., 2022; Zhang et al., 2023). Consequently, we further investigated and compared
the microscopic characteristics of soot particles during their transboundary transport
through the inland and through the sea pathways.

**3.2 Classification and fraction change of soot particles: inland vs. sea**
Based on morphology, components, and mixing states of individual transported
particles examined by TEM-EDS, they were classified into soot-containing, S-
OM/metal/fly ash/mineral, S-rich, and OM/metal/fly ash/mineral particles (Figure S5).
The specific classification criteria were described in Text S1. Figure S5 shows that the
number fraction of soot-containing particles in the NCP decreased from 45% and 51%
during the polluted period to 13% and 18% during the clean period following
transboundary transport of haze plumes through the inland and the sea pathways. When
large amounts of haze pollutants were transported into the YRD from the NCP through
the inland and the sea pathways, the number fraction of soot-containing particles in the
YRD increased from 38% and 34% during the clean period to 53% and 65% during the
polluted period (Figure S5). The change in the number fraction of soot-containing





particles in the NCP and the YRD during transboundary transport is consistent with the
variation of EC concentrations. These results suggest that abundant soot-containing
particles in the NCP were transported to the YRD following transboundary transport of
haze plumes.

The morphology and mixing states of soot particles can be changed during
transport due to atmospheric aging (Li et al., 2024). Figure 3 shows morphology of soot
particles and their mixing structures with other aerosol components observed by TEM.
Based on the mixing structure of soot particles, they were divided into three categories:
bare-like, partly-coated, and embedded soot particles (Figure 3). Bare-like soot
particles are characterized by being isolated and externally mixed with other aerosols
(Figure 3a). Partly-coated soot particles manifest as a portion of them being coated by
other aerosol components (Figure 3b). Embedded soot particles mean they are
completely enveloped by other aerosol materials (Figure 3c). Among these three types
of soot particles, bare-like soot particles were considered to be more freshly emitted,
while embedded soot particles were more aged (China et al., 2015). To observe
embedded soot particles more clearly, their non-refractory coatings (e.g., S-rich
particles) were sublimed under stronger electron beam (Figure 3c). In some individual
soot-containing particles, thin halos around aerosol components were observed (Figure
3c). These thin halos have been confirmed to be water rims left by the dehydrating of
aqueous particles because their EDS spectra are similar to the substrate but different
from the organic coating (Zhang et al., 2023). Therefore, soot aggregates with water
rims were identified as a type of embedded soot particles.

TEM observations showed that there were different numbers of soot cores in
individual soot-containing particles during transboundary transport (Figure 4a). Based
on the number of soot cores in individual soot-containing particles, we further divided
partly-coated soot-containing particles and embedded soot-containing particles into
them with 1 soot core, 2 soot cores, and ≥ 3 soot cores (Figure 4a). Figures 4b-c show
the variation in the number fraction of soot-containing particles with different mixing
structures and soot core numbers during transboundary transport through the inland and





the sea pathways. During the polluted period, partly-coated types were dominant in
soot-containing particles in the NCP, accounting for 62-67% (Figures 4b-c). Following
the transboundary transport of haze plumes through the inland pathway, the dominated
soot-containing particles changed from partly-coated at 67% by number in the NCP to
embedded types at 71% in the YRD (Figure 4b). Meanwhile, more than 75% of them
had one soot core (Figure 4b). However, we noticed that the soot core number in the
dominated soot-containing particles increased from 1 in the NCP to $\geq 3$ in the YRD in
addition to the change in the dominated mixing structures from partly-coated at 62%
by number to embedded ones at 72% when plentiful soot-containing particles were
transported through the sea pathway (Figure 4c). These results indicate that soot-
containing particles may be subject to different aging processes during their
transboundary transport through the inland and the sea pathways. Moreover, large
numbers of soot-containing particles with multiple soot aggregates were also observed
in an aged atmospheric environment (Wu et al., 2016). However, their aging
mechanisms were not effectively elucidated. The potential aging mechanisms for soot-
containing particles in two transboundary transport events are discussed in the
following section.

**3.3 Variation in microphysical characteristics of soot particles and potential aging**

**mechanisms: inland vs. sea**

Figure 5 shows number percentages of bare-like, partly-coated, and embedded
soot-containing particles with different numbers of soot cores in different size bins in
the NCP and the YRD during two transboundary transport events. Figure 6 displays
size distributions of partly-coated and embedded soot-containing particles during their
transboundary transport. Bare-like soot particles were mainly concentrated in the finer
size range of 0-200 nm during the transboundary transport (Figure 5). In the NCP,
partly-coated soot-containing particles with 1 soot core dominated soot-containing
particles and mainly distributed in the size range of 200-500 nm during the polluted
period (Figures 4b, c and 5a, c). Figure 6a, c shows consistent results that the size



distribution of partly-coated soot-containing particles in the NCP had a peak at 396 nm
for the transportation through the inland pathway and at 384 nm for the transportation
through the sea pathway. As embedded soot-containing particles became the dominant
type during the transboundary transport, their size distribution presented a peak at a
larger diameter of 505 nm (inland) and at a much larger diameter of 925 nm (sea)
compared to the former diameters at 464 nm and 446 nm (Figure 6). Meanwhile, the
preponderant soot-containing particles in the YRD, i.e., embedded ones with 1 core
(inland) and $\geq 3$ cores (sea), dominated in the coarser size range of 500-700 nm and in
the much coarser size range of > 1600 nm, respectively (Figure 5b, d). These findings
suggest that aging processes of soot-containing particles during the transboundary
transport through the sea pathway not only acquired more soot cores but also greatly
enlarged their sizes in contrast to the inland pathway. Consistently, high numbers of
soot cores were found in coarse particles of $\geq 800$ nm during transboundary transport
of biomass burning emissions (Chen et al., 2023).

The $D_p/D_c$ ratio of transboundary soot-containing particles was calculated to

reflect the coating thickness of soot particles and to quantify the aging degree of soot
particles (Figure 7). During two transboundary transport events, the mean $D_p/D_c$ ratios
of partly-coated and embedded soot-containing particles presented similar levels at
2.37-2.41 and 2.85-2.92 in the NCP (Figure 7). Following the transboundary transport
of soot-containing particles through the inland pathway, the mean $D_p/D_c$ ratios of partly-
coated and embedded soot-containing particles increased from 2.37 and 2.85 in the NCP
to 2.79 and 3.41 in the YRD (Figure 7a). This amount of increase for the $D_p/D_c$ ratio of
soot-containing particles is comparable to that from ~1.8 to ~2.2 during haze evolution
(Zhang et al., 2019b) and from 1.42 to 1.78 during dust storm transport (Xu et al., 2020).
Moreover, consistent with these studies, we observed a transition in the dominant
mixing state of soot particles with secondary coatings from partly-coated with single
soot core to embedded with single soot core configurations during the transboundary
transport through the inland pathway (Figure 4b), indicating that coagulation played a
negligible role in the aging process (China et al., 2015). Soot particles have been



demonstrated to promote the formation of secondary aerosols around them via
heterogeneous or aqueous-phase reactions (Farley et al., 2023; Han et al., 2013; Zhu et
al., 2025). Figure 8 displays mixing structures of soot-containing particles when they
invaded into the YRD through the inland and the sea pathways. It is noted that water
rims around soot-containing particles transported through the inland pathway were not
observed (Figure 8a, c-d). This implies that aqueous-phase chemistry contributed
minimally to secondary aerosol formation on soot particles during the transboundary
transport through the inland pathway. As a result, heterogeneous aging processes might
mainly drive the enhancement of secondary aerosols on soot-containing particles
transported through the inland pathway and enlarged their $D_p/D_c$ ratios. When soot-
containing particles were transported from the NCP to the YRD through the sea pathway,
the partly-coated $D_p/D_c$ ratio slightly increased from 2.41 to 2.66, but the embedded
$D_p/D_c$ ratio largely increased from 2.92 to 4.38 (Figure 7b). Similar results were also
found in cloud processes with the $D_p/D_c$ increase from 2.3 to 4.4 for embedded soot-
containing particles reported by Fu et al. (2022). Moreover, Xu et al. (2020) showed a
relatively high $D_p/D_c$ increase proportion of soot-containing particles at ~40% during
the transportation of dust storms from China across the East China Sea with humid air
to Japan. Based on observed and simulated RH in eastern China (Table S1 and Figures
1c-d), soot-containing particles could experience wetter environments with > 90% RH
during transboundary transport through the sea pathway compared with the inland
pathway. We indeed observed obvious water rims around soot-containing particles
transported through the sea pathway compared to the inland pathway (Figure 8). The
presence of water rims indicates that those soot-containing particles were in the aqueous
phase prior to being analyzed by TEM (Zhang et al., 2023). Liu et al. (2018) also
revealed pronounced aqueous-phase signatures surrounding cloud droplet residuals, as
indicated by water rims. AFM measurements further confirmed that the particles
transported through the sea pathway exhibited a droplet morphology (Figure S2b).
Moreover, the observed phenomenon of two or more soot cores within individual
particles transported through the sea pathway aligns with the findings that a single cloud



droplet can entrain numerous refractory aerosol particles (e.g., soot) (Ding et al., 2025; Liu et al., 2018). Therefore, soot-containing particles predominantly underwent cloud process aging under extremely high RH of > 90% conditions during the transboundary transport through the sea pathway, resulting in a significant thickening of coatings on soot cores.

The $D_f$ of soot particles serves as a critical metric for assessing their compactness and degree of atmospheric aging processes, providing a quantitative basis for black carbon characterization (Li et al., 2024; Pang et al., 2022). Figure 9 presents the evolution of $D_f$ for partly-coated and embedded soot particles during atmospheric transport through the inland and sea pathways. The $D_f$ of partly-coated and embedded soot particles increased from 1.81 and 1.90 in the NCP to 1.84 and 1.93 in the YRD following the transboundary transport through the inland pathway (Figures 9a-b), suggesting that secondary coatings formed via heterogeneous aging processes enhanced soot compactness during the transport. This varied result of soot $D_f$ during the transboundary transport through the inland pathway is similar to that during the dynamic progression of regional heavy haze pollution in winter (Zhang et al., 2023). However, when soot particles were transported to the YRD from the NCP through the sea pathway, their $D_f$ increased from 1.81 for partly-coated soot and 1.89 for embedded soot to 1.85 and 2.07 (Figures 9c-d). This suggests that the structural collapse of embedded soot particles was more pronounced compared to partly-coated soot particles during the transport through the sea pathway. Moreover, in contrast to the inland pathway, the $D_f$ of embedded soot particles transported through the sea pathway showed a 9.5% greater amplitude (Figure 9). This comparative result is consistent with the observed differences in the $D_p/D_c$ of soot-containing particles during two distinct atmospheric transport events (Figure 7), indicating that cloud process aging under extremely high RH of > 90% can greatly promote the structural collapse of soot aggregates. This mechanism can be ascribed to surface tension induced by the hygroscopic growth of secondary coatings on soot under elevated RH, which collapse the soot fractal morphology through water-mediated structural restructuring (Schnitzler



et al., 2017). Therefore, cloud process aging of soot-containing particles during the
transboundary transport through the sea pathway acted synergistically to (1) facilitate
the entrainment of multiple soot cores, (2) substantially enhance their $D_p/D_c$ ratios by
~50%, and (3) induce more pronounced collapse of soot fractal structures with $D_f$ from
1.89 to 2.07.

**3.4 Optical absorption of soot particles: inland vs. sea**


Based on mixing states of soot particles during the transboundary transport, the
light absorption enhancement ($E_{abs}$) of soot-containing particles with 1-3 cores and
different mixing structures (partly-coated and embedded configurations) was calculated
by the DDA combined with the EMBS. Considering that embedded soot cores were
often distributed at the periphery rather than the center within individual particles
(Figure 8), we conducted optical simulations of embedded soot-containing particles
based on this realistic mixing structure. In the optical calculation, the diameters of
single soot cores and coatings were presumed to 194 nm and 925 nm according to their
size distribution (Figures S6 and 6d), and the total volume of soot cores in individual
constructed particles was constant when their numbers were changed.
Figure 10a shows the change in the $E_{abs}$ of soot-containing particles following their
aging from partly-coated to embedded states. The $E_{abs}$ of soot-containing particles with
one soot core increased from 1.80 for the partly-coated structure to 2.83 for the
embedded structure (Figure 10a). When soot-containing particles had two soot cores,
the $E_{abs}$ increased from 1.74 to 2.44, representing a 0.4-fold increase, with soot aging
from partly-coated to embedded configurations (Figure 10a). Following the soot core
number increase to three, the $E_{abs}$ of soot-containing particles increased by 117% (from
1.04 to 2.26) when their mixing structures changed from partly-coated to embedded
status (Figure 10a). These results suggest that individual particles containing higher
numbers of soot cores demonstrate larger optical absorption amplification during
atmospheric aging processes although their $E_{abs}$ values were lower. Previous studies
also found lower absorption efficiency in cloud drops with higher numbers of soot cores




(Jacobson, 2006) and smaller $E_{abs}$ in simulated ambient particles with larger number
density of soot cores (Fierce et al., 2016). In addition, similar radiative absorption
changes for soot-containing particles with different numbers of soot cores were
observed during the transformation of soot core positions following the disappearance
of liquid-liquid phase separation between organic and inorganic components (Zhang et
al., 2022).
Based on the percentage, $D_p/D_c$, and $E_{abs}$ of soot-containing particles with different
mixing states and core numbers, we can compare the change in radiative absorption
capacity per unit the change in coating thicknesses of soot-containing particles during
the transboundary transport through the inland and the sea pathways. Figure 10b shows
$\Delta E_{abs}/\Delta(D_p/D_c)$ of transboundary soot-containing particles transported through the
inland and sea pathways. When soot-containing particles were transported from the
NCP to the YRD through the inland pathway, their $\Delta E_{abs}/\Delta(D_p/D_c)$ reached 0.6 (Figure
10b). However, the $\Delta E_{abs}/\Delta(D_p/D_c)$ of soot-containing particles was only 0.17 following
their transboundary transport through the sea pathway (Figure 10b). These findings
suggest that the radiative absorption amplification per unit $D_p/D_c$ change of
transboundary soot-containing particles reduced by 72% with the change in their
transport pathways from inland to sea. This can be ascribed to cloud processing during
the transboundary transport through the sea pathway inducing more soot cores within
single particles in contrast to the inland pathway, thereby reducing their optical
absorption, as shown in Figure 11. Beeler et al. (2024) also found consistent results that
much lower $E_{abs}$ variation for soot-containing particles with the thickening of coatings
in pyrocumulonimbus clouds compared to urban air. If embedded types in soot-
containing particles were presumed as the traditional core-shell model, the
$\Delta E_{abs}/\Delta(D_p/D_c)$ of transboundary soot-containing particles was extremely low at 0.01-
0.03 (Table S2). This result shows a large difference from the optical absorption
simulated with real mixing structures of soot-containing particles. Therefore, the
atmospheric humidity condition during the transport of soot particles not only affects
their aging processes but also influences their radiative absorption (Figure 11). In view





of that soot particles can be exposed to high-humidity atmospheric environments during
transboundary transport, climate models should incorporate multicore soot-containing
particles to refine current simulations of climate effects.

## 4. Conclusions and implications

Cold fronts triggered by the East Asian winter monsoon have frequently
transported substantial air pollutant loads from the NCP to downwind areas over 1000
kilometers away in recent years, significantly impacting the YRD region (Huang et al.,
2020; Zhao et al., 2021). To explore the variation in microphysical properties, mixing
states, and light absorption of soot particles in these haze pollutants and their aging
mechanisms during the transboundary transport, we conducted synchronized field
campaigns in December 2017 and December 2020 across the NCP and the YRD. Two
types of transboundary transport models (i.e., passing through the inland and the sea)
were identified based on transport pathways of haze plumes. According to the mixing
state of soot particles examined by TEM observations, they were divided into bare-like,
partly-coated, and embedded types. Meanwhile, the number of soot cores within
individual soot-containing particles was quantified.
Following the transboundary transport of haze pollutants through the inland
pathway, soot-containing particles underwent heterogeneous aging processes. This
aging process changed the dominated mixing state of soot-containing particles from
partly-coated types at 67% to embedded types at 71%, but the soot core number per
particle mainly remained at one. The median size and mean $D_p/D_c$ of partly-coated and
embedded soot-containing particles increased from 396-464 nm and 2.37-2.85 to 435-
505 nm and 2.79-3.41 during the transboundary transport through the inland pathway
because of secondary aerosol formation on soot particles via heterogeneous reactions.
In addition, the soot core $D_f$ increased from 1.81-1.90 to 1.84-1.93 under the
compacting effect of secondary coatings on soot aggregates. When soot-containing
particles were transported through the sea pathway, cloud process aging under
extremely high RH became their major evolution mechanisms. The cloud process aging



not only transformed the dominated soot-containing particles from partly-coated types at 62% to embedded types at 72% but also increased their soot core numbers from 1 to $\geq 3$. Compared to the inland pathway, the median size and mean $D_p/D_c$ of partly-coated soot-containing particles and their soot core $D_f$ showed similar variations during the transboundary transport through the sea pathway. However, these parameters for embedded soot-containing particles transported through the sea pathway represented larger increases from 446 nm, 2.92, and 1.89 in the NCP to 925 nm, 4.38, and 2.07 in the YRD.

Based on the optical simulation, transboundary soot-containing particles transported through the inland pathway exhibited a $\Delta E_{abs}/\Delta(D_p/D_c)$ of 0.6. Nevertheless, with the change in the transport pathway of soot-containing particles from the inland to the sea, the $\Delta E_{abs}/\Delta(D_p/D_c)$ reduced by 72% due to the entrainment of multiple soot cores by cloud processes. Our study demonstrates that soot particles, i.e., black carbon, undergo distinct evolutionary processes and exhibit altered microphysical and optical properties across different transport pathways. This necessitates incorporating meteorological conditions along transport pathways, particularly the elevated RH in sea pathways, into future assessments of black carbon optical properties. Given the scarce observational data on transboundary black carbon in the marine atmosphere compared to well-characterized those in the inland atmosphere, directly applying inland-based parameterization schemes to simulate optical properties of black carbon transported through sea pathways would introduce significant biases. Therefore, to accurately obtain optical properties of atmospheric transported black carbon, we suggest that future studies should prioritize multiscale characterization of black carbon mixing states and morphology in different transportation environments, particularly the cloud-processed mixing structure of multiple black carbon cores. Advanced single particle modeling, such as EMBS, that can reconstruct particles with real microphysical properties from TEM images could be coupled into macroscopic radiative forcing estimation (Wang et al., 2025). Ultimately, quantifying the climate impacts of black carbon necessitates a comprehensive understanding of how mixing state and



morphology evolution driven by atmospheric aging processes regulates absorption

enhancement to refine predictive models for climate mitigation strategies.



**Data availability**

All data presented in this paper are available upon request from the corresponding author (liweijun@zju.edu.cn).

**Author contributions**

JZ and WL conceived the study and wrote the manuscript. The field campaigns were organized and supervised by JZ and WL, and assisted by YW, LX, YZ, and HN. JZ, YW, LX, and YZ contributed the sample analyses. ZZ made the optical simulation. All authors reviewed and commented on the paper.

**Competing interests**

The authors declare that they have no conflict of interest.

**Acknowledgements**

This work was funded by the National Natural Science Foundation of China (42307141 and 42307143), Shandong Provincial Natural Science Foundation of China (ZR2023QD094 and ZR2023QD151), Zhejiang Provincial Natural Science Foundation of China (LZJMZ25D050002), and LAC/CMA (2023B10). We thanked Wenshuai Li for the sea level pressure simulation.



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



## Figure Captions

**Figure 1.** Meteorological fields in eastern China during the observation period. (a-b) Wind fields combined with surface PM$_{2.5}$ concentrations at 20:00 (local time) on December 30, 2017 and at 2:00 on December 8, 2020 derived from European Centre for Medium-Range Weather Forecasts (ECMWF, https://earth.nullschool.net/). The blue arrow dashed lines indicate prevailing wind direction. (c-d) Meteorological fields covering observation sites in the North China Plain (NCP) and Yangtze River Delta (YRD) at 1000 hpa.

**Figure 2.** Concentration-weighted trajectory (CWT) plots of PM$_{2.5}$ before arriving at observation sites in the YRD. (a-b) Nanjing and Hangzhou during December 30-31, 2017. (c-d) Nanjing and Hangzhou during December 7-8, 2020.

**Figure 3.** Typical transmission electron microscopy (TEM) images of soot particles in different mixing states. (a) Bare-like soot particle. (b) Partly-coated soot particles. (c) Embedded soot particles. Embedded soot particles in panel (c) can be clearly observed after their coatings are sublimed under strong electron beam.

**Figure 4.** Typical TEM images and number fractions of soot-containing particles with different mixing states and soot core numbers in two types of transboundary transport models from the NCP to the YRD. (a) Partly-coated and embedded soot-containing particles with different numbers of soot cores. (b) Variation in the number fraction of soot-containing particles during the transboundary transport through the inland pathway. (c) Variation in the number fraction of soot-containing particles during the transboundary transport through the sea pathway.

**Figure 5.** Number fractions of soot-containing particles with different mixing states and numbers of soot cores in different size bins in two types of transboundary transport models from the NCP to the YRD. (a-b) Soot-containing particles transported through the inland pathway. (c-d) Soot-containing particles transported through the sea pathway.

**Figure 6.** Number size distribution of soot-containing particles in two types of transboundary transport models from the NCP to the YRD. (a-b) Size distribution of soot-containing particles transported through the inland pathway. (c-d) Size distribution



of soot-containing particles transported through the sea pathway.
**Figure 7.** The size ratio of soot-containing particles to their soot cores ($D_p/D_c$) in two
types of transboundary transport models from the NCP to the YRD. (a) $D_p/D_c$ ratios of
soot-containing particles transported through the inland pathway. (b) $D_p/D_c$ ratios of
soot-containing particles transported through the sea pathway. A schematic model of
the $D_p/D_c$ ratio of soot-containing particles with the core-shell structure is exampled.
**Figure 8.** Low magnification TEM images of soot-containing particles in the YRD
during two transboundary transport. (a) Soot-containing particles transported through
the inland pathway. (b) Soot-containing particles transported through the sea pathway.
(c-d) Magnified TEM images for soot-containing particles in panel (a). (e-f) Magnified
TEM images for soot-containing particles in panel (b).
**Figure 9.** Variation in the fractal dimension ($D_f$) of partly-coated and embedded soot
particles during their transboundary transport from the NCP to the YRD. (a-b) $D_f$ of
soot particles transported through the inland pathway. (c-d) $D_f$ of soot particles
transported through the sea pathway. A schematic model of the soot $D_f$ is exemplified.
**Figure 10.** Variation in the optical absorption of soot-containing particles. (a) The light
absorption enhancement ($E_{abs}$) of partly-coated and embedded soot-containing particle
models relative to their soot cores. (b) The change in $E_{abs}$ per unit the change in $D_p/D_c$
($\Delta E_{abs}/\Delta(D_p/D_c)$) of soot-containing particles during two transboundary transport
events through the inland and the sea pathways. Partly-coated and embedded soot-
containing particle models constructed by the Electron-Microscope-to-BC-Simulation
(EMBS) tool were exampled in panel (a).
**Figure 11.** A schematic diagram for the change in the mixing state and optical
absorption of soot-containing particles during the transboundary transport from the
NCP to the YRD through the inland and the sea pathways. (a) Soot-containing particles
undergo heterogeneous aging processes during the transboundary transport through the
inland pathway, which mainly change their mixing states from partly-coated with single
soot core to embedded with single soot core structures and increase the $E_{abs}$ change per
unit $D_p/D_c$ change at 0.6. (b) Following the transboundary transport of soot-containing



particles through the sea pathway, cloud process aging becomes the dominated
evolution mechanism of soot-containing particles. This process not only transforms the
mixing state of soot-containing particles from partly-coated with single soot core to
embedded with multiple soot core structures but also slightly enhances the $E_{abs}$ change
per unit $D_p/D_c$ change at 0.17.

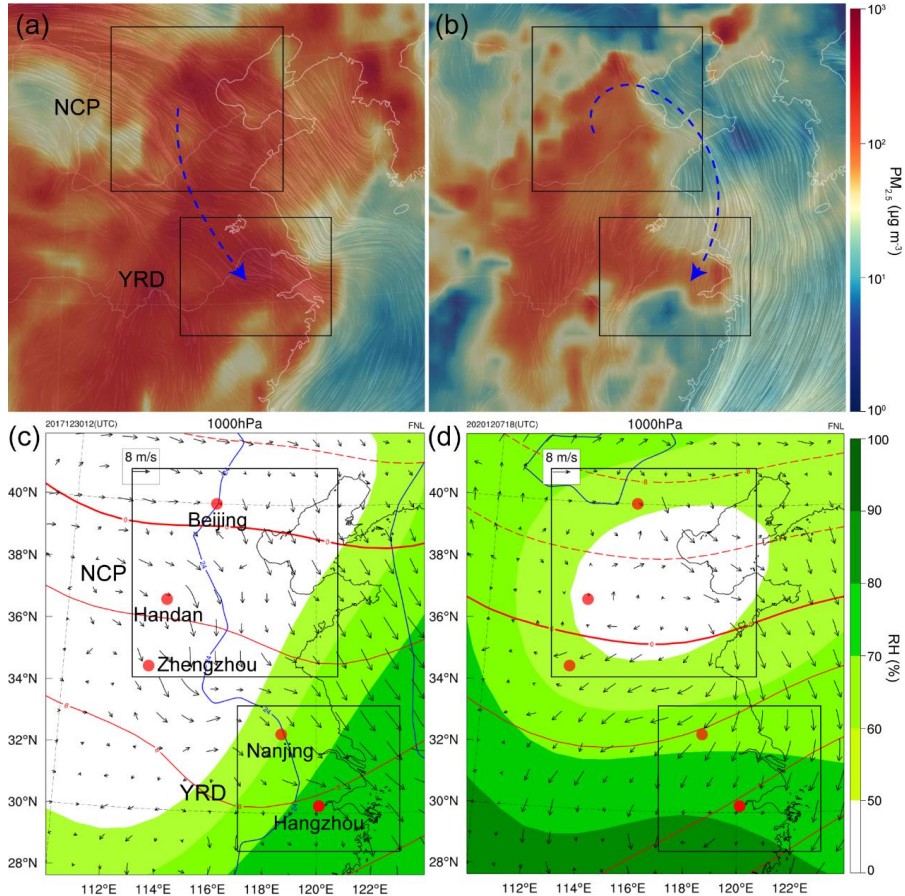

**Figure 1.** Meteorological fields in eastern China during the observation period. (a-b)
Wind fields combined with surface PM$_{2.5}$ concentrations at 20:00 (local time) on
December 30, 2017 and at 2:00 on December 8, 2020 derived from European Centre
for Medium-Range Weather Forecasts (ECMWF, https://earth.nullschool.net/). The
blue arrow dashed lines indicate prevailing wind direction. (c-d) Meteorological fields
covering observation sites in the North China Plain (NCP) and Yangtze River Delta
(YRD) at 1000 hpa.



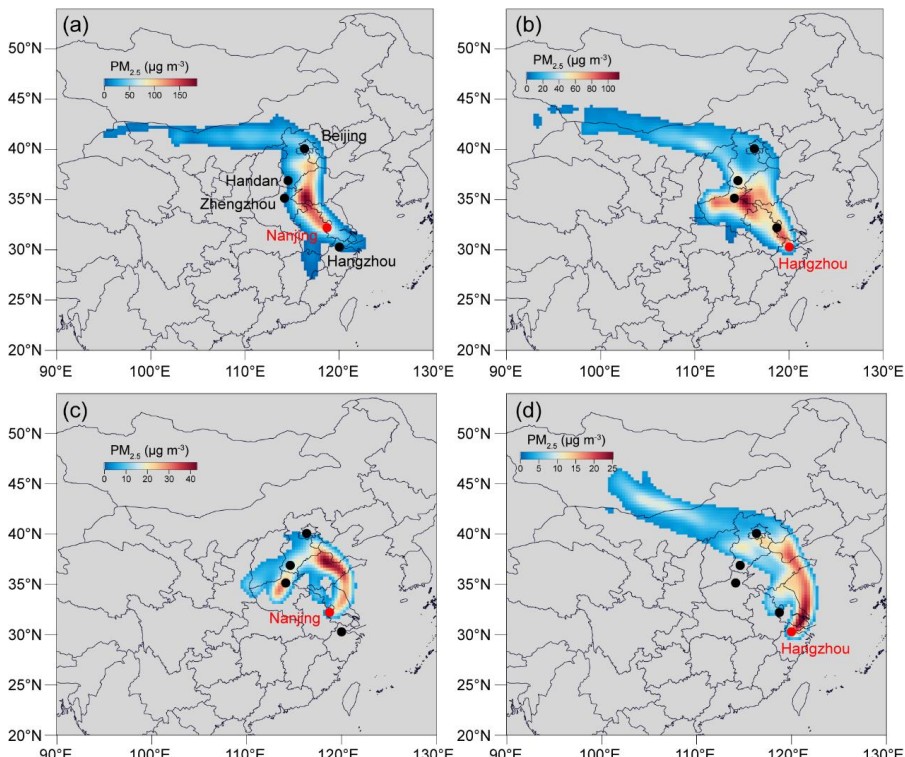

**Figure 2.** Concentration-weighted trajectory (CWT) plots of PM$_{2.5}$ before arriving at observation sites in the YRD. (a-b) Nanjing and Hangzhou during December 30-31, 2017. (c-d) Nanjing and Hangzhou during December 7-8, 2020.



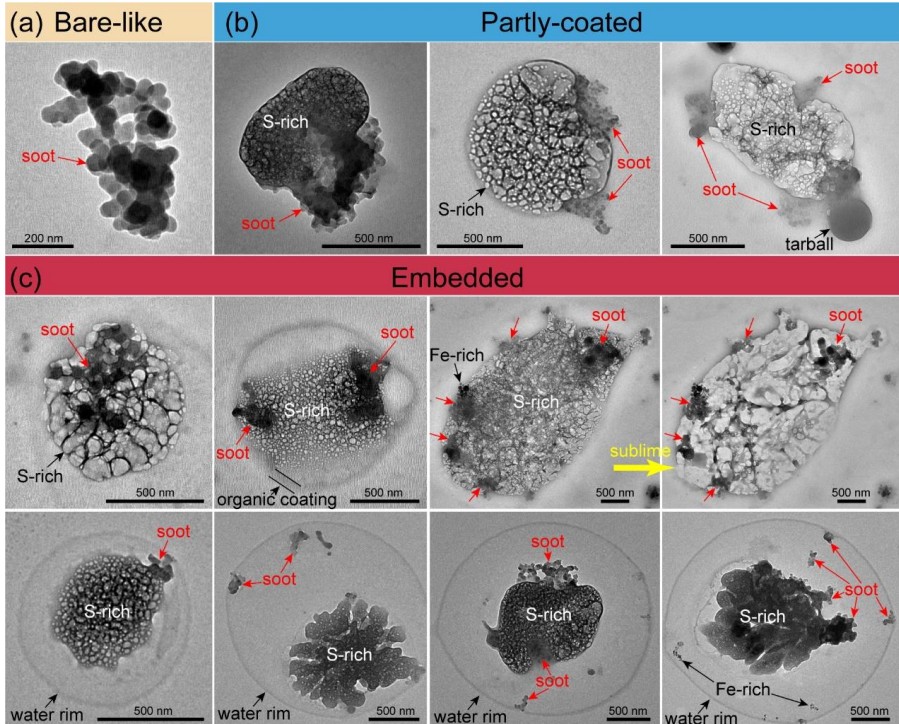

**Figure 3.** Typical transmission electron microscopy (TEM) images of soot particles in different mixing states. (a) Bare-like soot particle. (b) Partly-coated soot particles. (c) Embedded soot particles. Embedded soot particles in panel (c) can be clearly observed after their coatings are sublimed under strong electron beam.

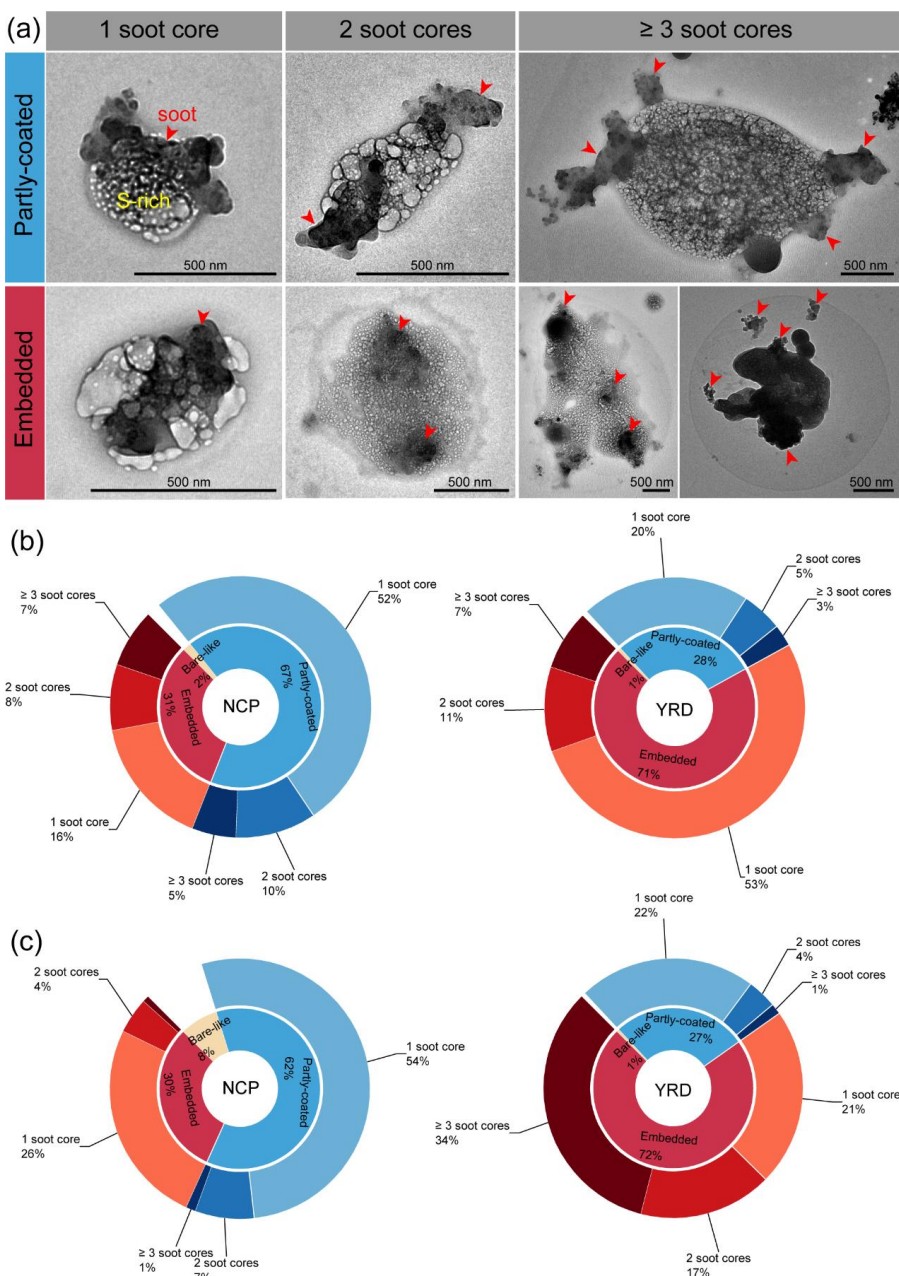

**Figure 4.** Typical TEM images and number fractions of soot-containing particles with different mixing states and soot core numbers in two types of transboundary transport models from the NCP to the YRD. (a) Partly-coated and embedded soot-containing particles with different numbers of soot cores. (b) Variation in the number fraction of



soot-containing particles during the transboundary transport through the inland pathway.

(c) Variation in the number fraction of soot-containing particles during the

transboundary transport through the sea pathway.

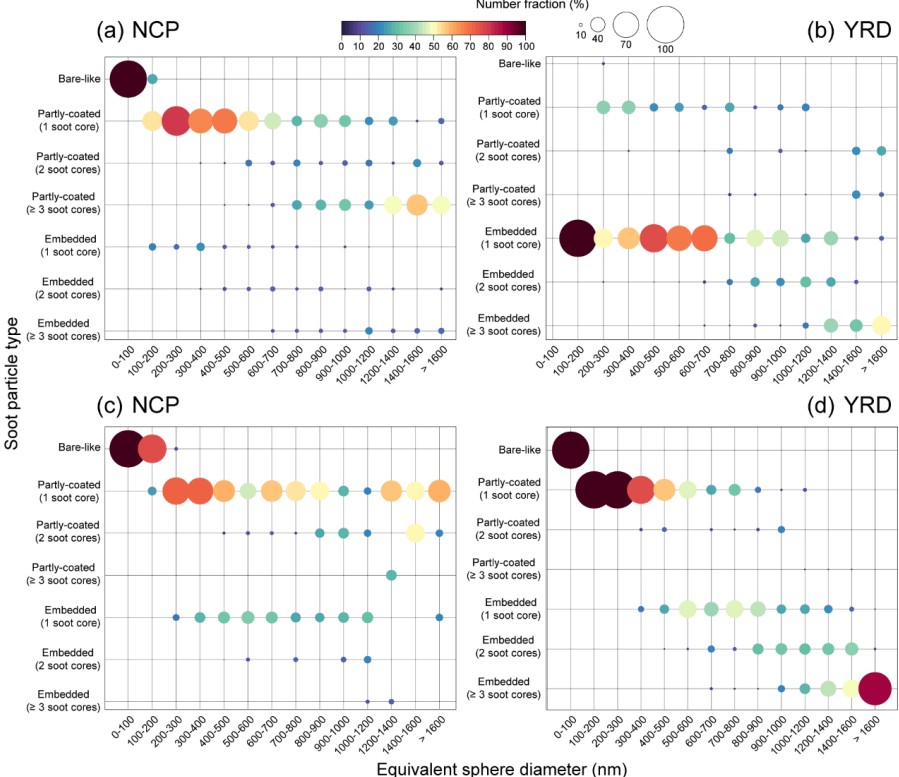

**Figure 5.** Number fractions of soot-containing particles with different mixing states and numbers of soot cores in different size bins in two types of transboundary transport models from the NCP to the YRD. (a-b) Soot-containing particles transported through the inland pathway. (c-d) Soot-containing particles transported through the sea pathway.

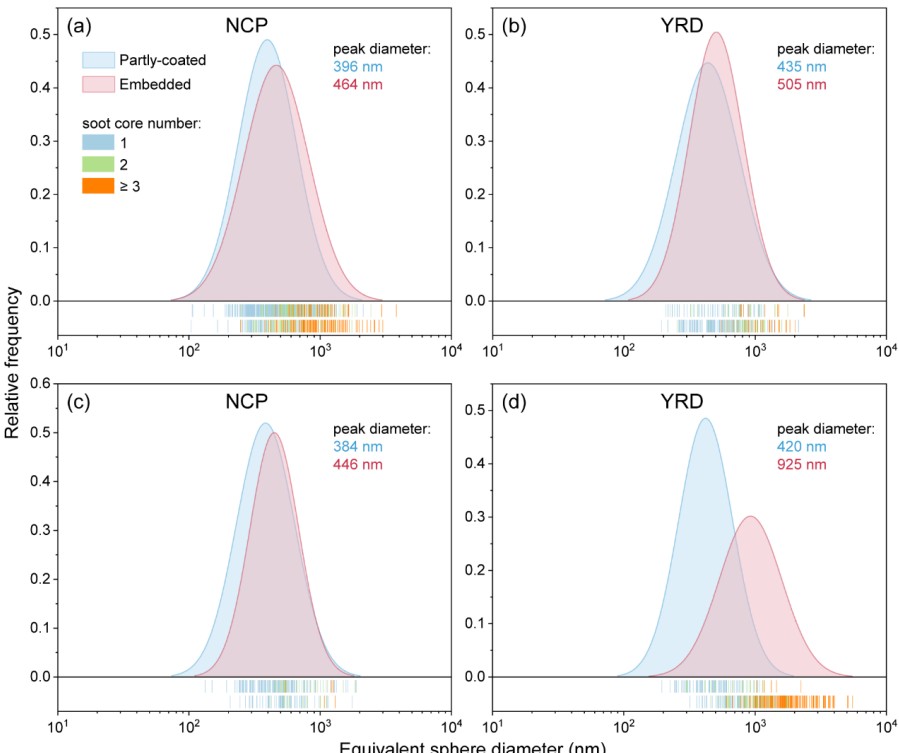

**Figure 6.** Number size distribution of soot-containing particles in two types of transboundary transport models from the NCP to the YRD. (a-b) Size distribution of soot-containing particles transported through the inland pathway. (c-d) Size distribution of soot-containing particles transported through the sea pathway.

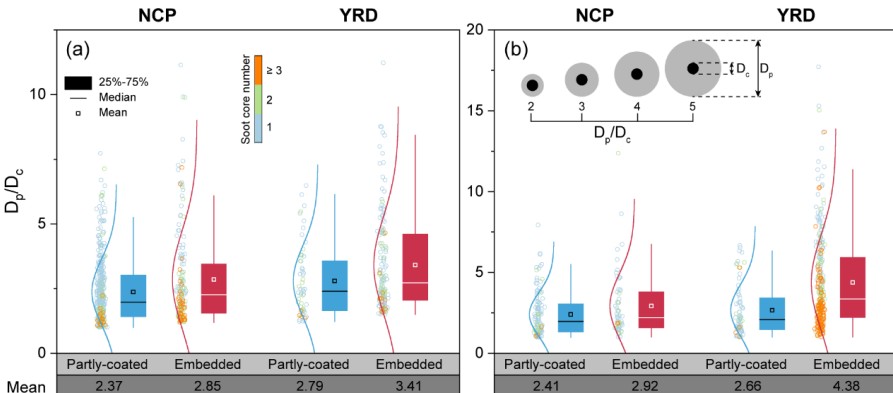

**Figure 7.** The size ratio of soot-containing particles to their soot cores ($D_p/D_c$) in two types of transboundary transport models from the NCP to the YRD. (a) $D_p/D_c$ ratios of soot-containing particles transported through the inland pathway. (b) $D_p/D_c$ ratios of soot-containing particles transported through the sea pathway. A schematic model of the $D_p/D_c$ ratio of soot-containing particles with the core-shell structure is exampled.

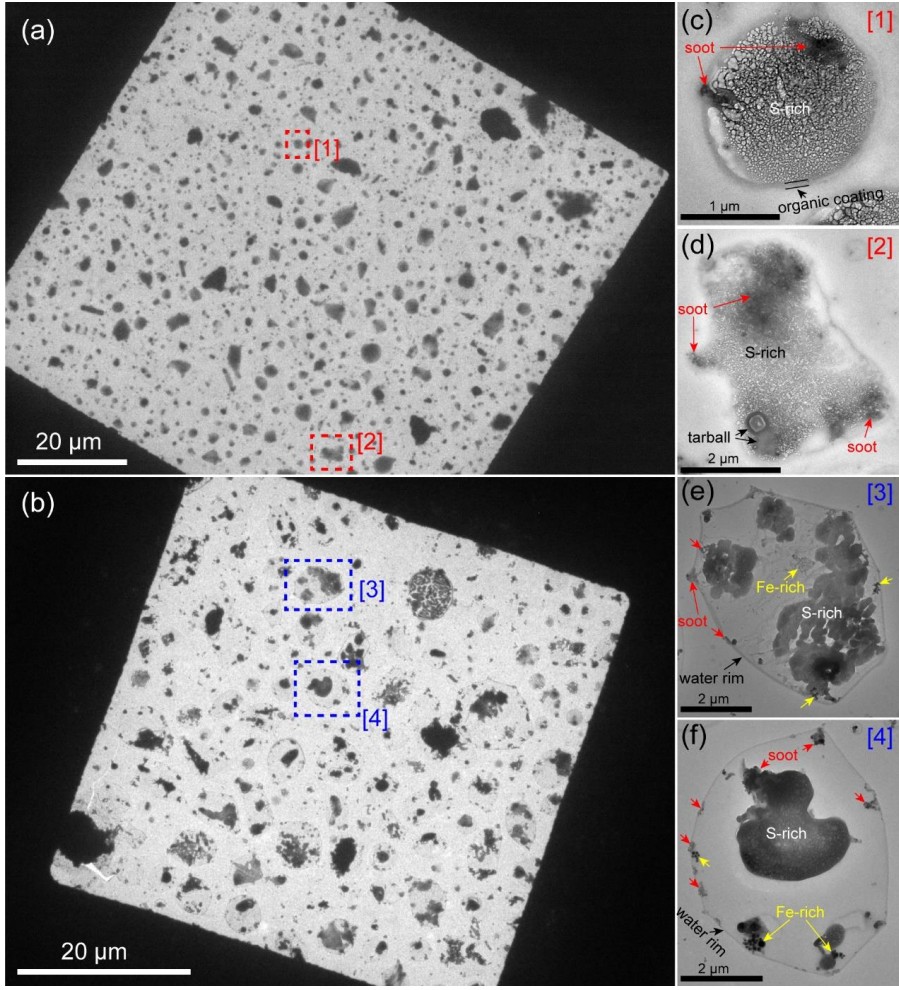

**Figure 8.** Low magnification TEM images of soot-containing particles in the YRD during two transboundary transport. (a) Soot-containing particles transported through the inland pathway. (b) Soot-containing particles transported through the sea pathway. (c-d) Magnified TEM images for soot-containing particles in panel (a). (e-f) Magnified TEM images for soot-containing particles in panel (b).

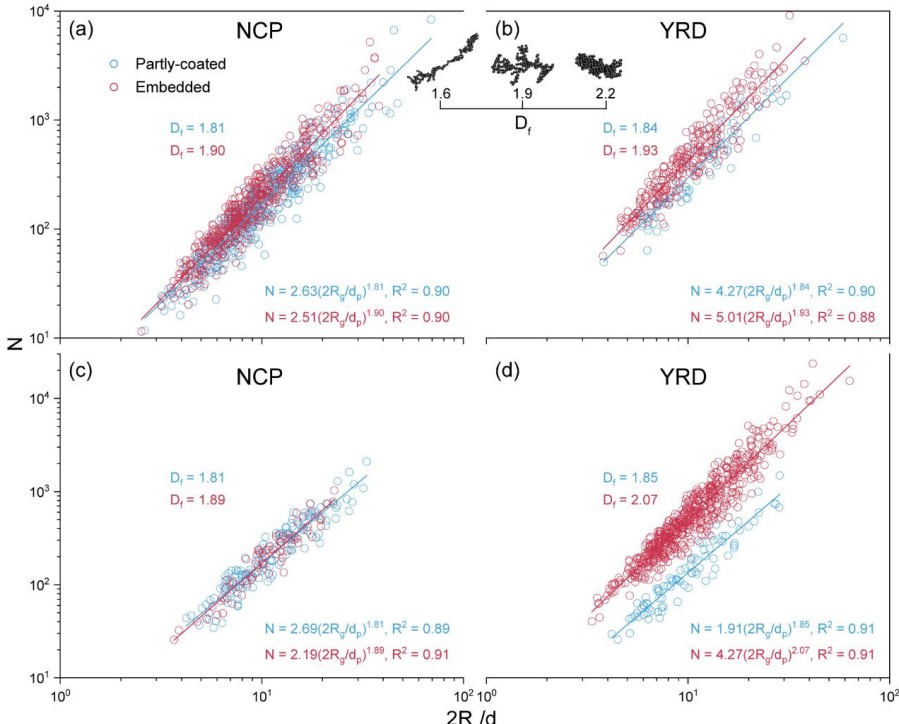

**Figure 9.** Variation in the fractal dimension ($D_f$) of partly-coated and embedded soot particles during their transboundary transport from the NCP to the YRD. (a-b) $D_f$ of soot particles transported through the inland pathway. (c-d) $D_f$ of soot particles transported through the sea pathway. A schematic model of the soot $D_f$ is exemplified.

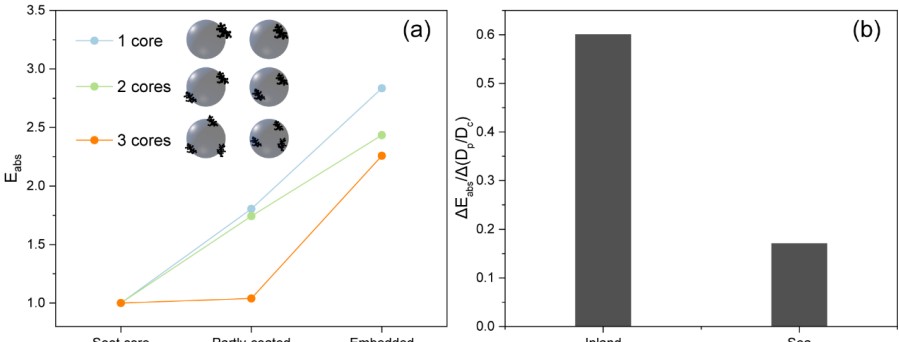

**Figure 10.** Variation in the optical absorption of soot-containing particles. (a) The light absorption enhancement ($E_{abs}$) of partly-coated and embedded soot-containing particle models relative to their soot cores. (b) The change in $E_{abs}$ per unit the change in $D_p/D_c$ ($\Delta E_{abs}/\Delta(D_p/D_c)$) of soot-containing particles during two transboundary transport events through the inland and the sea pathways. Partly-coated and embedded soot-containing particle models constructed by the Electron-Microscope-to-BC-Simulation (EMBS) tool were exampled in panel (a).





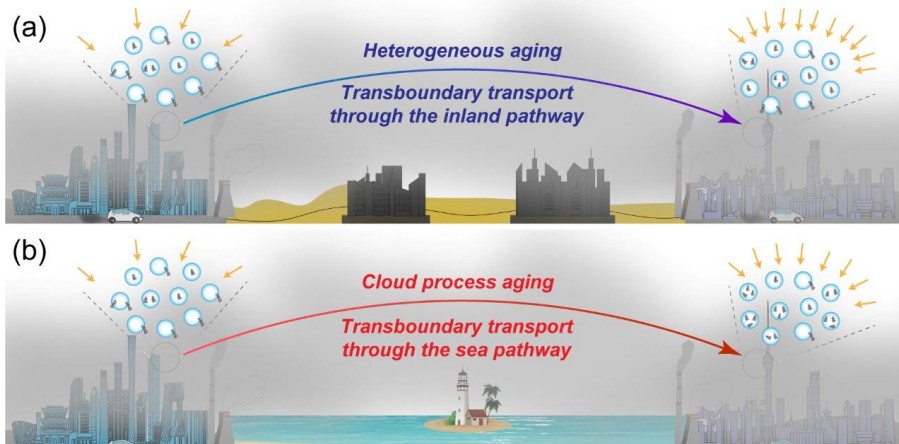

**Figure 11.** A schematic diagram for the change in the mixing state and optical absorption of soot-containing particles during the transboundary transport from the NCP to the YRD through the inland and the sea pathways. (a) Soot-containing particles undergo heterogeneous aging processes during the transboundary transport through the inland pathway, which mainly change their mixing states from partly-coated with single soot core to embedded with single soot core structures and increase the $E_{abs}$ change per unit $D_p/D_c$ change at 0.6. (b) Following the transboundary transport of soot-containing particles through the sea pathway, cloud process aging becomes the dominated evolution mechanism of soot-containing particles. This process not only transforms the mixing state of soot-containing particles from partly-coated with single soot core to embedded with multiple soot core structures but also slightly enhances the $E_{abs}$ change per unit $D_p/D_c$ change at 0.17.