# Peer review of "Disparate evolution mechanisms and optical"

_EGUsphere, 2025_

## Author Comment (AC1)

**Reply to Reviewer #1**

This is an interesting work, based on the observation of different patterns in aerosols associated with different wind trajectories. However, some clarifications are needed. Section 3.4 about optical absorption should be revised. Although English is good in general, grammar should be revised. Notation is not always defined or is not self-consistent. Structure (main manuscript/SM) should also be revised.

**We are grateful for this reviewer's comments. These comments are all valuable and helpful for improving our paper.**

**As the reviewer's comments, we moved some descriptions into the Supplementary Materials (Text S1). Moreover, we answered the comments carefully and made corrections in the submitted manuscript and supplementary information. The corrections and the responses are as following:**

**In the revised manuscript and supplementary information, the red color was marked as the revised places.**

1. Segregation of information into the Supplementary Material should be done when this information is not essential for understanding the study. In this case, the information in the SM is essential to follow the main manuscript. Authors are suggested to reorganize the information.

   **Reply: We appreciated the reviewer's comments and moved some descriptions into the Supplementary Materials (Text S1).**

   **P11 L264-267: "Based on the association between $PM_{2.5}$ concentrations and prevailing winds described in Text S1, we inferred that there was a typical transboundary transport process of pollutants from the NCP to the YRD on December 30-31, 2017 and December 7-8, 2020, respectively."**

2. Introduction: This reviewer does not agree with some conventions often used in environmental articles, such as the equivalence between black carbon and elemental carbon (BC is a carbonaceous combustion-derived aerosol, while elemental carbon is the major chemical component of BC, but also of any organic

material), or the list of soot sources (why fossil fuels and biomass??; any liquid biofuel, electrofuel, or non-biologic waste material will also emit soot when burned).

**Reply: Thanks for the reviewer's comments and we considered them carefully. We revised this sentence as follows.**

**P4 L96-98: "Soot particles dominated by black carbon, mainly emitted from incomplete burning of fossil, biomass, and other non-biological fuels, are important light absorbing aerosols in fine particles (Bond et al., 2013)."**

3. Introduction: This reviewer does not agree that "simulating soot climate effect is readily achievable in models", as stated. The variety of sizes, shapes, compositions, and nanostructures affect the optical properties of soot and makes the simulation very difficult.

**Reply: Thanks. To eliminate the misunderstanding, we revised this sentence as follows.**

**P5 L115-118: "When simulating soot climate effect in models, the complicated microphysical properties of soot particles could be underestimated due to limited studies, thereby introducing considerable uncertainties into the results (Chen et al., 2025; IPCC, 2021)."**

4. Section 2.2: If OM/OC ratios in Chinese cities is assumed as 1.91, what organic matter is the remaining 0.91/1.91? Why

**Reply: In this study, the OM/OC mass ratio is assumed as 1.91. OC (organic carbon) refers to the total carbon mass in organic compounds. OM (organic matter) refers to the organic compound mass. Thus, the remaining 0.91 refers to the total mass of other elements (e.g., O, H, N, and S) in organic compounds.**

**We revised this sentence as follows.**

**P7 L168-170: "Considering the contribution of other elements (e.g., O, H, N, and S) to the mass of organic matter (OM, i.e., organic compound), OM**

**concentrations were obtained by multiplying organic carbon (OC) concentrations by 1.91 reported by Xing et al. (2013).”**

5. Sections 2.2 and 3.2: TEM operates under high vacuum. Therefore, evaporation or sublimation of coatings could occur even in "conventional TEM observations". What is the change in the beam power to distinguish between "enhanced electron beam observations" and "conventional observations", and thus between enhanced and conventional absorption? Could authors include TEM images of the same particle before and after enhancing the beam power? Visible bubbles observed in Figures 3 and 4, indicating evaporation, are declared to correspond after enhancing power, but how do these particles look like before? Are diameters "Dp" those obtained with TEM under conventional mode and "Dc" those obtained under enhanced electron beam? Please clarify.

**Reply: As the reviewer commented, some coatings could be sublimated under conventional TEM observations. However, the sublimation rate is relatively slow because these coatings usually consist of sulfate and nitrate mixtures, which maintain their original morphology longer than pure ammonium nitrate. Moreover, we used a weak electron beam during conventional TEM observations, which minimized damage to particles and mitigated their sublimation. We also obtained TEM images within five seconds.**

**To present the morphology of the indiscernible soot more clearly, we enhanced the electron beam via adjusting the beam spot size to accelerate the sublimation of non-refractory coatings. Figure 3c shows TEM images of the same particle before and after exposure to the enhanced electron beam. The TEM images in Figures 3 and 4 were all obtained before the electron beam was increased, except for the sublimated one. The bubbles in some coatings are obvious, which may be related to their chemical compositions and the imaging duration. In this study, only particles containing indiscernible soot were re-imaged under high electron beam conditions. The**

$D_p$ and discernible soot $D_c$ were measured under conventional TEM observations, while the indiscernible soot $D_c$ values were measured under TEM observations with enhanced electron beam.

**P7 L181-182: "These parameters for indiscernible soot particles were measured under TEM observations with the enhanced electron beam."**

**P13-14 L348-350: "To observe some indiscernible embedded soot particles more clearly, their non-refractory coatings (e.g., S-rich particles) were sublimed under an enhanced electron beam (Figure 3c)."**

[Figure]

**Figure 3.** Typical transmission electron microscopy (TEM) images of soot particles in different mixing structures. (a) Bare-like soot particle. (b) Partly-coated soot particles. (c) Embedded soot particles. Some indiscernible embedded soot particles in panel (c) can be clearly observed after their coatings are sublimed under an enhanced electron beam.

**6.** Sections 2.2 and 3.2: What do authors exactly mean by "mixing states" of soot

particles? Is it an appropriate name? Based on Text S1, it seems that they refer to chemical composition. However, based on Section 3.2 and Figures 3 and 4, it seems that they refer to bare-like, partly-coated or embedded. Please correct or clarify.

**Reply: The "mixing state" refers to one type of aerosol particles (e.g., soot) mixing with other aerosols (e.g., S-rich particles). Mixing structure represents morphological mixing between two different types of aerosol particles, including bare-like, partly-coated, and embedded. To emphasize the mixing characteristics of soot with other aerosols, we modified "mixing states of soot particles" to "mixing structures of soot particles".**

**P7 L175-177: "To better observe soot mixing structures and measure soot geometrical parameters, we enhanced the electron beam to sublime non-refractory coatings of indiscernible soot cores after conventional TEM observations."**

**P13 L338-339: "The morphology and mixing structures of soot particles can be changed during transport due to atmospheric aging (Li et al., 2024)."**

7. Section 2.2: Equations 4 to 7 are written without a brief explanation of their meaning. Authors should at least explain that ignoring the overlap (or sintering or interpenetration) between monomers would lead to underestimation of the fractal prefactor of the power-law relationship (eq. 4). Moreover, publications after 1997 have demonstrated that also this prefactor (not only that in eq. 5) is highly affected by the overlap parameter (see, e.g., Powder Technology 271, 141–154 (2015)).

**Reply: Thanks. We added some explanations for equations 4 to 7.**

**P8 L207-208: "The monomer number in soot particles and the gyration radius of soot particles can be calculated using the following equations."**

**P9 L212-213: "Because the fractal prefactor is highly affected by the overlap between soot monomers (Lapuerta et al., 2015), an overlap parameter needs to be considered."**

8. Section 2.3: Parameters "n" and "W" in equation 9 are not defined. Please check uniformity in the notation.

**Reply: We added the definition of parameters "n" and "W" in equation 9.**

**P10 L238-239: "where W is weighting factor; n is the number of all trajectory endpoints in a grid cell; log(n+1) represents the density of trajectories."**

9. Section 3.1: In Figure S4 the content in EC (supposedly associated with soot) is very minor (purple). On the contrary, in Figure S5, the percent of soot-containing particles is very high (light blue). How do these results match?

**Reply: As shown in Figure 7, the average $D_p/D_c$ value of soot-containing particles is ~3. Based the $D_p/D_c$ value, we can calculate the volume ratio of soot-containing particles to soot cores at ~27. Moreover, soot sizes are mainly lower than 200 nm (Figures 5 and S8). These suggest that the total volume of soot is very low. As we known, the soot density is also low. Thus, the total mass of soot is very minor although the soot number is large. These results derived from individual particle analysis and bulk analysis are matched.**

[Figure]

**Figure 7.** The size ratio of soot-containing particles to their soot cores ($D_p/D_c$) in two types of transboundary transport models from the NCP to the YRD. (a) $D_p/D_c$ ratios of soot-containing particles transported through the inland pathway. (b) $D_p/D_c$ ratios of soot-containing particles transported through the sea pathway. A schematic model of the $D_p/D_c$ ratio of soot-containing particles with the core-shell structure is exampled.

[Figure]

**Figure 5.** Number fractions of soot-containing particles with different mixing structures and numbers of soot cores in different size bins in two types of transboundary transport models from the NCP to the YRD. (a-b) Soot-containing particles transported through the inland pathway. (c-d) Soot-containing particles transported through the sea pathway.

[Figure]

**Figure S8.** Size distribution of soot cores in individual soot-containing particles.

**10.** Section 3.3: What is, in the opinion of authors, the dominant reason for the increase in the size, the number of soot cores, the Dp/Dc ratio, and the fractal dimension of soot structures: coalescence between agglomerates (entrainment) or breakage of agglomerates inside the aerosol (collapse)?

**Reply: Based on the results and discussion in section 3.3, heterogeneous aging processes and cloud aging processes of soot-containing particles mainly lead to the increases in their sizes, $D_p/D_c$ ratios, and $D_f$ during the transport through the inland pathway and the sea pathway, respectively. Only cloud aging processes can cause the increase in soot core numbers. Also, one recent study observed the similar phenomenon (Chen et al., 2025). No matter what kind of aging process, we believe that the $D_f$ increase is caused by the structural collapse of soot cores.**

**11.** Section 3.4: This reviewer can understand that the energy adsorbed is reduced for the sea pathway with respect to the inland pathway, and even that multiple cores may also contribute to reduce absorption. But does not understand why the energy absorbed from embedded particles is higher than that absorbed from soot cores (Figure 10a). The refractive index of soot is much higher than that of coatings (and specially its imaginary part, related to attenuation of light). Consequently, the ageing process should lead to a decrease in the energy

absorbed. Please, revise, or explain better.

**Reply: Thanks for the reviewer's comments. The higher light absorption of embedded particles compared to soot cores can be attributed to the lensing effect. The lensing effect refers to that when soot particles age in the atmosphere and form a core-shell structure, their coatings (e.g., sulfates and nitrates) refract and focus light like a lens, enhancing the absorption capacity of soot particles to solar radiation (Cappa et al., 2012; Fierce et al., 2020). At present, many studies have found the lensing effect of soot particles (Liu et al., 2015; Riemer et al., 2019; Wang et al., 2025). We added some explanations for the lensing effect.**

**P20 L533-536: "Figure 10a shows the change in the $E_{abs}$ of soot-containing particles following their aging from bare-like to partly-coated, and then to embedded states. Compared to soot cores, partly-coated and embedded soot-containing particles present higher $E_{abs}$ (Figure 10a), due to the lensing effect (Fierce et al., 2020; Wang et al., 2025)."**

12. Technical corrections: please correct "If the high-pressure system located" to "If the high-pressure system is located"; "Obviously, there was a bench of data" to "Obviously, there is a bench of data"; "However, transboundary haze pollutants crossed the East China Sea remain unexplored" to "However, transboundary haze pollutants crossing the East China Sea remain unexplored".

**Reply: Thanks. We revised them.**

**P4 L82-85: "If the high-pressure system is located in the interior of the NCP, heavy haze covering the Jing-Jin-Ji region (i.e., Beijing, Tianjin, and Hebei) could move out from inland China to the East China Sea and return into the inland region under prevailing winds, influencing air quality of the YRD (see section 3.1)."**

**P4 L90-91: "Obviously, there is a bench of data available from national ground monitoring net station of air quality to support the measurements and modelling studies."**

**P4 L92-93: "However, transboundary haze pollutants crossing the East China Sea remain unexplored."**

**References**

Bond, T. C., Doherty, S. J., Fahey, D. W., Forster, P. M., Berntsen, T., DeAngelo, B. J., Flanner, M. G., Ghan, S., Kärcher, B., Koch, D., Kinne, S., Kondo, Y., Quinn, P. K., Sarofim, M. C., Schultz, M. G., Schulz, M., Venkataraman, C., Zhang, H., Zhang, S., Bellouin, N., Guttikunda, S. K., Hopke, P. K., Jacobson, M. Z., Kaiser, J. W., Klimont, Z., Lohmann, U., Schwarz, J. P., Shindell, D., Storelvmo, T., Warren, S. G., and Zender, C. S.: Bounding the role of black carbon in the climate system: A scientific assessment, J. Geophys. Res.-Atmos., 118, 5380-5552, https://doi.org/10.1002/jgrd.50171, 2013.

Cappa, C. D., Onasch, T. B., Massoli, P., Worsnop, D. R., Bates, T. S., Cross, E. S., Davidovits, P., Hakala, J., Hayden, K. L., Jobson, B. T., Kolesar, K. R., Lack, D. A., Lerner, B. M., Li, S. M., Mellon, D., Nuaaman, I., Olfert, J. S., Petaja, T., Quinn, P. K., Song, C., Subramanian, R., Williams, E. J., and Zaveri, R. A.: Radiative Absorption Enhancements Due to the Mixing State of Atmospheric Black Carbon, Science, 337, 1078-1081, https://doi.org/10.1126/science.1223447, 2012.

Chen, X., Ching, J., Wu, F., Matsui, H., Jacobson, M. Z., Zhang, F., Wang, Y., Zhang, Z., Liu, D., Zhu, S., Rudich, Y., Shi, Z., Yoo, H., Jeon, K.-J., and Li, W.: Locating the missing absorption enhancement due to multi–core black carbon aerosols, Nat. Commun., 16, 10187, https://doi.org/10.1038/s41467-025-65079-2, 2025.

Fierce, L., Onasch, T. B., Cappa, C. D., Mazzoleni, C., China, S., Bhandari, J., Davidovits, P., Fischer, D. A., Helgestad, T., Lambe, A. T., Sedlacek, A. J., Smith, G. D., and Wolff, L.: Radiative absorption enhancements by black carbon controlled by particle-to-particle heterogeneity in composition, Proc. Natl. Acad. Sci. U. S. A., 117, 5196-5203, https://doi.org/10.1073/pnas.1919723117, 2020.

IPCC: Climate Change 2021: the Physical Science Basis: Contribution of Working Group I to the Sixth Assessment Report of the Intergovernmental Panel on Climate Change (IPCC). 2021.

Lapuerta, M., Expósito, J. J., and Martos, F. J.: Effect of sintering on the fractal prefactor of agglomerates, Powder Technol., 271, 141-154, https://doi.org/10.1016/j.powtec.2014.10.041, 2015.

Li, W., Riemer, N., Xu, L., Wang, Y., Adachi, K., Shi, Z., Zhang, D., Zheng, Z., and Laskin, A.: Microphysical properties of atmospheric soot and organic particles: measurements, modeling, and impacts, npj Clim. Atmos. Sci., 7, 65, https://doi.org/10.1038/s41612-024-00610-8, 2024.

Liu, S., Aiken, A. C., Gorkowski, K., Dubey, M. K., Cappa, C. D., Williams, L. R., Herndon, S. C., Massoli, P., Fortner, E. C., Chhabra, P. S., Brooks, W. A., Onasch, T. B., Jayne, J. T., Worsnop, D. R., China, S., Sharma, N., Mazzoleni, C., Xu, L., Ng, N. L., Liu, D., Allan, J. D., Lee, J. D., Fleming, Z. L., Mohr, C., Zotter, P., Szidat, S., and Prévôt, A. S. H.: Enhanced light absorption by mixed source black and brown carbon particles in UK winter, Nat. Commun., 6, 8435, https://doi.org/10.1038/ncomms9435, 2015.

Riemer, N., Ault, A. P., West, M., Craig, R. L., and Curtis, J. H.: Aerosol Mixing State: Measurements, Modeling, and Impacts, Rev. Geophys., 57, 187-249, https://doi.org/https://doi.org/10.1029/2018RG000615, 2019.

Wang, Y., Zheng, Z., Sun, Y., Yao, Y., Ma, P.-L., Zhang, A., Zhu, S., Zhang, Z., Chen, X., Pang, Y., Wang, Q., Che, H., Ching, J., and Li, W.: Improved representation of black carbon mixing structures suggests stronger direct radiative heating, One Earth, 8, https://doi.org/10.1016/j.oneear.2025.101311, 2025.

Xing, L., Fu, T. M., Cao, J. J., Lee, S. C., Wang, G. H., Ho, K. F., Cheng, M. C., You, C. F., and Wang, T. J.: Seasonal and spatial variability of the OM/OC mass ratios and high regional correlation between oxalic acid and zinc in Chinese urban organic aerosols, Atmos. Chem. Phys., 13, 4307-4318, https://doi.org/10.5194/acp-13-4307-2013, 2013.

---

## Author Comment (AC2)

**Reply to Reviewer #2**

The manuscript by Zhang et al. investigates two different soot aerosol aging processes by comparing the properties of particles that are transported over land and over the ocean. The authors utilize measurements of particle morphology, number of soot cores and coating thickness, and report differences in optical properties. Overall, it was found that particles that were transported via the sea pathway, and likely underwent aqueous-phase or cloud processing, had an increase in soot core number and a decrease in absorption enhancement compared to particles that were transported over the land, where heterogenous oxidation dominated.

Overall, the study is well conducted, clear, and of interest to the ACP community. I would recommend publication after addressing a few minor comments.

**We are grateful for this reviewer's comments. These comments are all valuable and helpful for improving our paper. We added a figure to show the presence of clouds during the transport of haze masses through the sea pathway (Figure S7). We added the *P* value calculation to show the significant difference in coating thicknesses of soot-containing particles between the NCP and the YRD (Figure 7). We added labels (Inland and Sea) in Figure 2, 4, 5, 6, 7, 8, 9. We answered the comments carefully and have made corrections in the submitted manuscript. The corrections and the responses are as following:**

**In the revised manuscript and supplementary information, the red color was marked as the revised places.**

1. The authors primarily discuss environmental conditions (cloud processing, RH, etc) as what is causing the differences between the two aging pathways. Although this evidence is convincing, there are likely also significantly different emissions mixing with the haze plume during transport over land than over the ocean. This is briefly mentioned as a possibility at line 95 but not referred back to in the results section. Can the authors comment on if these differences and if they have implications on the observations or results.

   **Reply: We appreciated the reviewer's comments and added some discussion**

on differences in pollutant emissions and their implications.

P18 L482-487: "We further noticed that soot-containing particles did not pass areas with high emissions during transboundary transport through the sea pathway compared to those transported through the inland pathway (Figures 1a-b). These findings suggest that the aging process of soot-containing particles was primarily driven by the meteorological change (i.e., cloud), with minimal contribution from additional industrial and urban emissions along the sea pathway."

[Figure]

**Figure 1.** Meteorological fields in eastern China during the observation period. (a-b) Wind fields combined with surface $PM_{2.5}$ concentrations at 20:00 (local time) on December 30, 2017 and at 2:00 on December 8, 2020 derived from European Centre for Medium-Range Weather Forecasts (ECMWF, https://earth.nullschool.net/). The

blue arrow dashed lines indicate prevailing wind direction. (c-d) Meteorological fields covering observation sites in the North China Plain (NCP) and Yangtze River Delta (YRD) at 1000 hpa.

2. Line 327: Figure 3 and 4 show particles with soot cores very close to the particle edge. How exactly are partly-coated and embedded particles differentiated. Is there a specific threshold for how much soot is exposed for it to be considered partly-coated? Similarly, how did the authors categorize multi-core particles where individual cores were embedded and partly-coated in the same particle.

**Reply: In this study, soot cores completely coated by S-rich particles were considered to be embedded types and other internally mixed soot cores were considered to be partly-coated types.**

**Based on TEM observations, less than 10% of soot-containing particles had both embedded and partly-coated soot cores. To categorize these particles, we classified those with more than 95% of the total soot volume embedded in host particles as embedded soot-containing particles, and the remainder as partly-coated types. Because these particles were relatively few, they had a limited impact on the statistical results. We added the explanation as follows.**

P14 L360-365: "In this study, less than 10% of soot-containing particles had both embedded and partly-coated soot cores. To categorize these particles, we classified those with more than 95% of the total soot volume embedded in host particles as embedded soot-containing particles, and the remainder as partly-coated types. Because these particles were relatively few, they had a limited impact on the statistical results."

3. Line 392: Are the differences in coating thickness between NCP and YRD statistically significant?

**Reply: Yes, the differences in coating thicknesses of soot-containing particles between the NCP and the YRD are significant. We calculated the *P* value to show it (Figure 7). The weak difference in partly-coated soot-containing**

particles transported through the sea pathway may be attributed to the lack of additional pollutants involved in their aging processes and their lack of activation.

P16 L423-426: "Following the transboundary transport of soot-containing particles through the inland pathway, the mean $D_p/D_c$ ratios of partly-coated and embedded soot-containing particles increased from 2.37 ± 1.27 and 2.85 ± 1.89 in the NCP to 2.79 ± 1.37 and 3.41 ± 1.87 in the YRD ($P < 0.05$, Figure 7a)."

P17 L446-449: "When soot-containing particles were transported from the NCP to the YRD through the sea pathway, the partly-coated $D_p/D_c$ ratio slightly increased from 2.41 ± 1.37 to 2.66 ± 1.58, but the embedded $D_p/D_c$ ratio significantly increased from 2.92 ± 2.01 to 4.38 ± 2.92 ($P < 0.001$, Figure 7b)."

[Figure]

**Figure 7.** The size ratio of soot-containing particles to their soot cores ($D_p/D_c$) in two types of transboundary transport models from the NCP to the YRD. (a) $D_p/D_c$ ratios of soot-containing particles transported through the inland pathway. (b) $D_p/D_c$ ratios of soot-containing particles transported through the sea pathway. A schematic model of the $D_p/D_c$ ratio of soot-containing particles with the core-shell structure is exampled.

**4.** Can the authors expand on their discussion of the water rim observed in some particles, as this seems to be an important piece of evidence for aqueous phase processing (i.e. line 427). It would be helpful to clarify if this is a marker for particles that have undergone aqueous-phase processing (as mentioned at line 431), or for particles that contained an aqueous-phase when analyzed with TEM (line 429). Put another way: If the particles underwent aqueous-phase processing during transport, then were subjected to lower humidity conditions and effloresced, would the rim still be present.

**Reply: Yes, the water rim will still exist after aqueous-phase particle efflorescence. The previous studies have observed water rims after aqueous-phase particles are dehydrated using an individual particle hygroscopicity system (Sun et al., 2018). We added a discussion on the water rim.**

**P17 L462-465: "Laboratory studies have also observed water rims after aqueous-phase particles are dehydrated (Sun et al., 2018). In other words, if particles undergo aqueous-phase processing during transport, and then effloresce under low RH conditions, the water rim will be present as a marker."**

**5.** Line 437: Is there satellite or meteorological evidence of clouds being present along the back trajectory of the airmass. If possible, It would be useful to differentiate between cloud processing and high humidity (but still subsaturated)

**Reply: Thanks for the reviewer's comments. We added a satellite image combined with the backward trajectory to reflect the presence of clouds during the transport of haze masses through the sea pathway (Figure S7).**

**P18 L475-478: "Figure S7 shows the satellite image combined with the backward trajectory of haze masses during December 7-8, 2020. We found the presence of clouds over the East China Sea during the transport of haze masses through the sea pathway (Figure S7)."**

[Figure]

**Figure S7.** A satellite image combined with the backward trajectory of haze masses before arriving at Nanjing and Hangzhou sites during December 7-8, 2020.

**6.** Line 512: Can the authors clarify why entrainment of multiple soot cores results in lower ΔEabs/Δ(Dp/Dc).

**Reply: Yes, the lower $\Delta E_{abs}/\Delta(D_p/D_c)$ of soot-containing particles transported through the sea pathway may be attributed to the larger change in their $D_p/D_c$ and smaller change in their $E_{abs}$ compared with the inland pathway. We added some discussion on it.**

**P21 L560-572: "Previous studies have revealed that the $E_{abs}$ of soot-containing particles first increases and then tends to stabilize with their coating thickness (e.g., $D_p/D_c$) increases (Beeler et al., 2024; Fu et al., 2022). We found that the mean $D_p/D_c$ of embedded soot-containing particles exhibited a large value at 4.38 when haze masses were transported through the sea pathway (Figure 7b). In addition, cloud processes induced multiple soot cores within single particles during the transboundary transport through the sea pathway in contrast to the inland pathway, reducing their optical absorption (Figure 10a). Beeler et al. (2024) also found consistent results that much lower $E_{abs}$ variation for soot-containing particles with the thickening of coatings in pyrocumulonimbus clouds compared with urban air. Therefore, the larger $D_p/D_c$ change and the smaller $E_{abs}$ change of soot-containing particles transported through the sea**

**pathway should result in the lower $\Delta E_{abs}/\Delta(D_p/D_c)$ compared to those transported through the inland pathway."**

[Figure]

**Figure 10.** Variation in the optical absorption of soot-containing particles. (a) The light absorption enhancement ($E_{abs}$) of partly-coated and embedded soot-containing particle models relative to their soot cores. (b) The change in $E_{abs}$ per unit the change in $D_p/D_c$ ($\Delta E_{abs}/\Delta(D_p/D_c)$) of soot-containing particles during two transboundary transport events through the inland and the sea pathways. Partly-coated and embedded soot-containing particle models constructed by the Electron-Microscope-to-BC-Simulation (EMBS) tool were exampled in panel (a).

7. Most figures (Figure 2, 4, 5, 6, 7, 8, 9) have panels for the results of each pathway, however it is not immediately clear which one is which. Although this information is in the caption, I would recommend labeling the row or panels with the different pathways.

   **Reply: Thanks. We added labels (Inland and Sea) in these Figures.**

[Figure]

**Figure 2.** Concentration-weighted trajectory (CWT) plots of PM$_{2.5}$ before arriving at observation sites (Nanjing and Hangzhou) in the YRD. (a-b) Transboundary transport through the inland pathway during December 30-31, 2017. (c-d) Transboundary transport through the sea pathway during December 7-8, 2020.

[Figure]

**Figure 4.** Typical TEM images and number fractions of soot-containing particles with different mixing structures and soot core numbers in two types of transboundary transport models from the NCP to the YRD. (a) Partly-coated and embedded soot-containing particles with different numbers of soot cores. (b)

Variation in the number fraction of soot-containing particles during the transboundary transport through the inland pathway. (c) Variation in the number fraction of soot-containing particles during the transboundary transport through the sea pathway.

[Figure]

**Figure 5.** Number fractions of soot-containing particles with different mixing structures and numbers of soot cores in different size bins in two types of transboundary transport models from the NCP to the YRD. (a-b) Soot-containing particles transported through the inland pathway. (c-d) Soot-containing particles transported through the sea pathway.

[Figure]

**Figure 6.** Number size distribution of soot-containing particles in two types of transboundary transport models from the NCP to the YRD. (a-b) Size distribution of soot-containing particles transported through the inland pathway. (c-d) Size distribution of soot-containing particles transported through the sea pathway.

[Figure]

**Figure 8.** Low magnification TEM images of soot-containing particles in the YRD during two transboundary transport. (a) Soot-containing particles transported through the inland pathway. (b) Soot-containing particles transported through the sea pathway. (c-d) Magnified TEM images for soot-containing particles in panel (a). (e-f) Magnified TEM images for soot-containing particles in panel (b).

[Figure]

**Figure 9.** Variation in the fractal dimension ($D_f$) of partly-coated and embedded soot particles during their transboundary transport from the NCP to the YRD. (a-b) $D_f$ of soot particles transported through the inland pathway. (c-d) $D_f$ of soot particles transported through the sea pathway. A schematic model of the soot $D_f$ is exemplified.

8. Line 193: When calculating Dp/Dc, how are multiple soot cores handled. Is ESDsoot the sum of all soot cores?

   **Reply: When we calculated the $D_p/D_c$ of soot-containing particles with multiple soot cores, the total volume of soot cores was first computed. Then, based on the total volume, we calculated the diameter of the soot core and the $D_p/D_c$ of soot-containing particles with multiple soot cores.**

9. Line 278: It would be helpful for context to include an average (or range of) transport time based on the back trajectories.

**Reply: We added the simulated time for backward trajectories.**

**P11 L285-287: "To determine whether the transport pathway of pollutants was consistent with the wind field, the PM$_{2.5}$ transport pathway was simulated based on the 72 hr CWT analysis (Figure 2)."**

10. Line 98: "…exerting favorable effects on global warming in the atmosphere". I would recommend changing from "favorable" to "positive radiative forcing" or similar.

    **Reply: We revised it as follows.**

    **P4 L98-99: "They exert positive radiative forcing effects on global warming in the atmosphere (Cappa et al., 2012; Jacobson, 2001)."**

11. Line 293: "These results suggest that massive primary and secondary aerosols including EC (i.e., soot) were transported from the NCP to the YRD under cold fronts, both through the inland and the sea pathways". The word "massive" here is confusing. I would recommend changing it unless the authors are referring to the size of the particles.

    **Reply: Thanks, we revised this word.**

    **P12 L303-305: "These results suggest that many primary and secondary aerosols including EC were transported from the NCP to the YRD under cold fronts, both through the inland and the sea pathways."**

**References**

Beeler, P., Kumar, J., Schwarz, J. P., Adachi, K., Fierce, L., Perring, A. E., Katich, J. M., and Chakrabarty, R. K.: Light absorption enhancement of black carbon in a pyrocumulonimbus cloud, Nat. Commun., 15, 6243, https://doi.org/10.1038/s41467-024-50070-0, 2024.

Cappa, C. D., Onasch, T. B., Massoli, P., Worsnop, D. R., Bates, T. S., Cross, E. S., Davidovits, P., Hakala, J., Hayden, K. L., Jobson, B. T., Kolesar, K. R., Lack, D. A., Lerner, B. M., Li, S. M., Mellon, D., Nuaaman, I., Olfert, J. S., Petaja, T., Quinn, P. K., Song, C., Subramanian, R., Williams, E. J., and Zaveri, R. A.: Radiative Absorption Enhancements Due to the Mixing State of Atmospheric Black Carbon, Science, 337, 1078-1081, https://doi.org/10.1126/science.1223447, 2012.

Fu, Y., Peng, X., Sun, W., Hu, X., Wang, D., Yang, Y., Guo, Z., Wang, Y., Zhang, G., Zhu, J., Ou, J., Shi, Z., Wang, X., and Bi, X.: Impact of Cloud Process in the Mixing State and Microphysical Properties of Soot Particles: Implications in Light Absorption Enhancement, J. Geophys. Res.-Atmos., 127, e2022JD037169, https://doi.org/10.1029/2022JD037169, 2022.

Jacobson, M. Z.: Strong radiative heating due to the mixing state of black carbon in atmospheric aerosols, Nature, 409, 695-697, https://doi.org/10.1038/35055518, 2001.

Sun, J., Liu, L., Xu, L., Wang, Y., Wu, Z., Hu, M., Shi, Z., Li, Y., Zhang, X., Chen, J., and Li, W.: Key Role of Nitrate in Phase Transitions of Urban Particles: Implications of Important Reactive Surfaces for Secondary Aerosol Formation, Journal of Geophysical Research: Atmospheres, 123, 1234-1243, https://doi.org/10.1002/2017JD027264, 2018.

---

## Author Comment (AC3)

**Reply to Reviewer #3**

Understanding how aerosol particle age during transport is critically important for improving model performance yet still suffers from the paucity of field data that can guide model development. The current manuscript by Zhang et al., provides much needed property and aging data on $PM_{2.5}$ particles by being able to compare the pathway impacts over inland vs the sea. To carry out this study, the authors examined the microphysical and optical properties of haze particles from the North China Plain (NCP) to the Yangtze River Delta (YRD) where cold fronts exhibited two different pathway: an inland path and a "sea" path that took the air mass across the East China Sea. The authors report that the "dryish" inland path favors a heterogeneous-centric aging pathway with single-core embedded soot, whereas the sea pathway, with its concomitantly very high RH and cloud processing favors entrainment and coalescence resulting in the production of super micron particles characterized by multiple soot cores inside large droplets/residuals that exhibit a lower absorption enhancement than that calculated for the inland pathway particles. This manuscript tackles an important problem with a modern, multi-method approach and yields novel, physically meaningful insights about how transport environment affects soot aging and absorption. This manuscript is recommended for publication AFTER some moderate revisions that are focused on clarifying the uncertainties and representativeness in the measurements, greatly increasing the language transparency in discussing the cloud process aging as the dominate mechanism vs. other high RH processes and increasing the narrative on the underlying modeling assumptions. These are discussed below. Also, the narrative is convoluted at several points within the manuscript and thus the authors are encouraged to have the manuscript grammar reviewed.

**We are grateful for this reviewer's comments. These comments are all valuable and helpful for improving our paper. We added a figure to indicate the representativeness of the transboundary transport event through the sea pathway (Figure S4). We added the standard deviation of the number fraction, $D_p/D_c$, and $D_f$ in Figure 4, 7, 9. We also added the significant difference of $D_p/D_c$ of soot-containing particles during two transport events and made the scales on**

the ordinates same in Figure 7. We added a satellite image to indicate the presence of clouds along the sea transport pathway (Figure S7). We calculated the volume ratio of soot coatings to soot cores ($V_{coating}/V_c$) in Table S2. We added a sensitivity test to show the credibility of 72% $\Delta E_{abs}/\Delta(D_p/D_c)$ reduction (Figure S9). The manuscript grammar was also reviewed. Moreover, we answered the comments carefully and have made corrections in the submitted manuscript and supplementary information. The corrections and the responses are as following:

In the revised manuscript and supplementary information, the red color was marked as the revised places.

**1.** The analysis rests primarily on two winter haze events (2017, 2020). These events appear well chosen and documented, but they still represent a limited sample of meteorological regimes. How representative are these case studies and to what extent are they generalizable to other years, and source composition (e.g., wildfire aerosols vs fossil fuel)? It is assumed that the transport trajectories indicate that the haze plumes stayed in the boundary layer and did not punch through to the free troposphere.

**Reply: We appreciated the reviewer's comments. Our previous studies investigated some transboundary transport haze events from the NCP to the YRD across inland areas (Zhang et al., 2023; Zhang et al., 2021). Their meteorological fields were similar to those of the transboundary transport event through the inland pathway in this study, that is strong northerly winds over eastern China under the influence of the Siberian cold high. In recent years, we have found that as the position of the high-pressure system shifts, haze aerosols from the NCP can be transported to the YRD through the sea pathway. Such transport patterns were observed not only in the second event but also during other time periods (Figure S4). Previously, only a limited number of modeling studies had identified this transport route (Wu et al., 2022). Moreover, we found that the chemical composition of the transported haze aerosols was similar in several transport events, with**

secondary inorganic ions and organic matter being the dominant components. This is consistent with previous studies on other transboundary transport haze events from the NCP to the YRD (Huang et al., 2020; Li et al., 2019; Xie et al., 2023). Therefore, we can confirm that these two events are representative cases of transboundary transport of haze pollutants from the NCP to the YRD through the inland and the sea pathways. We added representative explanations.

**P11 L275-277: "This meteorological field was similar to those of transboundary transport haze events from the NCP to the YRD across inland areas (Hou et al., 2020; Hu et al., 2021; Zhang et al., 2023)."**

**P11 L281-282: "Such wind patterns were observed not only in the second event but also in other periods, as shown in Figure S4."**

**P12 L305-309: "This is consistent with previous studies on the transboundary transport of haze aerosols from the NCP to the YRD (Huang et al., 2020; Li et al., 2019; Xie et al., 2023). In summary, we can confirm that these two events represent typical cases of transboundary transport of haze pollutants from the NCP to the YRD through the inland and the sea pathways."**

[Figure]

**Figure S4.** Wind fields combined with surface PM$_{2.5}$ concentrations in eastern China. (a) 11:00 (local time) on January 4, 2020. (b) 4:00 (local time) on February 8, 2021.

2. While the Reviewer appreciates the effort of analyzing 3642 particles, the authors need to ascribe uncertainties/confidence intervals on all reported fractions (e.g., 62 vs. 67%; 71 vs. 72%) and on Dp/Dc and Df. It is also suggested that a statistical significance test be performed in the comparison between inland and sea cases.

**Reply: Thanks for the reviewer's comments. We added the standard deviation of the number fraction, $D_p/D_c$, and $D_f$ in Figure 4, 7, 9 and the manuscript. We also added the significant difference of $D_p/D_c$ of soot-containing particles during two transport events in Figure 7.**

[Figure]

**Figure 4.** Typical TEM images and number fractions of soot-containing particles with different mixing structures and soot core numbers in two types of transboundary transport models from the NCP to the YRD. (a) Partly-coated and embedded soot-containing particles with different numbers of soot cores. (b)

Variation in the number fraction of soot-containing particles during the transboundary transport through the inland pathway. (c) Variation in the number fraction of soot-containing particles during the transboundary transport through the sea pathway.

[Figure]

**Figure 7.** The size ratio of soot-containing particles to their soot cores ($D_p/D_c$) in two types of transboundary transport models from the NCP to the YRD. (a) $D_p/D_c$ ratios of soot-containing particles transported through the inland pathway. (b) $D_p/D_c$ ratios of soot-containing particles transported through the sea pathway. A schematic model of the $D_p/D_c$ ratio of soot-containing particles with the core-shell structure is exampled.

[Figure]

**Figure 9.** Variation in the fractal dimension ($D_f$) of partly-coated and embedded soot particles during their transboundary transport from the NCP to the YRD. (a-b) $D_f$ of soot particles transported through the inland pathway. (c-d) $D_f$ of soot particles transported through the sea pathway. A schematic model of the soot $D_f$ is exemplified.

3. While the observation of water rims and droplet-like morphologies are compelling indicators of aqueous processing, they do not uniquely prove that cloud processing is indeed the route. One could observe similar water rims from deliquesced aerosol under very high RH below cloud. Indeed, the latter only requires a very high RH condition, similar to that reported, whereas the former requires supersaturation conditions for cloud droplet formation, which is not discussed in the manuscript. Thus, the authors need to clarify the distinction between in-cloud vs. sub-cloud aqueous processing as opposed to simply citing "cloud process". Similarly, while it is reasonable to assumption that multiple

cores (2 - 3) per particle are primarily due to cloud entrainment and collision-coalescence, other pathways (e.g., coagulation in a wet aerosol layer, co-injection and growth of BC-rich, organic-rich droplets) could contribute. Therefore, the authors are encouraged to provide a narrative (argument) that simple coagulation of soot-containing particles in a high-RH boundary layer is insufficient to produce the observed size and multicore distributions thereby supporting the proposed cloud droplet pathway. Similarly, the authors are encouraged to be more tempered with their statement (page 20; lines 550 - 551) that "cloud process aging under extremely high RH became them major evolution mechanisms." An extremely high RH environment does not necessarily mean a cloud droplet environment (supersaturation conditions).

**Reply: Thanks. As the reviewer commented, the water rim and droplet-like morphology are not direct evidence of soot-containing particles undergoing cloud processes. These phenomena can also be observed under non-supersaturated high RH conditions. Furthermore, we agree with the reviewer's comment that other physicochemical pathways could also lead to multicore states in soot-containing particles, similar to cloud processes. Therefore, we have carefully considered the reviewer's suggestion by adding a discussion on the variation of particle size distribution. We also emphasized that simple coagulation of soot-containing particles in high RH environments was insufficient to explain the observed particle sizes and multicore characteristics, supporting the role of cloud processing. Additionally, we added a satellite image to further indicate the presence of clouds along the transport pathway (Figure S7). The sentence in lines 550-551 has also been revised as suggested.**

**P18 L468-478: "It is noted that the peak diameter and core number of embedded soot-containing particles largely shifted from 446 nm and 1 to 925 nm and ≥ 3 during transboundary transport through the sea pathway (Figures 4c and 6c-d). The evolution implies that simple coagulation or condensation was not the primary aging processes of soot-containing**

**particles in high RH environments, because these mechanisms are insufficient to explain the observed micron-sized particles with multiple cores (Liu et al., 2018). Instead, cloud processing likely played a more important role. Figure S7 shows the satellite image combined with the backward trajectory of haze masses during December 7-8, 2020. We found the presence of clouds over the East China Sea during the transport of haze masses through the sea pathway (Figure S7)."**

**P23 L623-625: "When soot-containing particles were transported through the sea pathway, cloud process aging became their major evolution mechanisms."**

[Figure]

**Figure 6.** Number size distribution of soot-containing particles in two types of transboundary transport models from the NCP to the YRD. (a-b) Size distribution of soot-containing particles transported through the inland pathway. (c-d) Size

distribution of soot-containing particles transported through the sea pathway.

[Figure]

**Figure S7.** A satellite image combined with the backward trajectory of haze masses before arriving at Nanjing and Hangzhou sites during December 7-8, 2020.

**4.** While Dp/Dc is conventional for submicron size particles with a single-core BC, it is not the most robust choice for the super-micron, cloud-processed, multicore particles that dominate this study. In this size regime that is the focus of this study, Dp and Dc are both equivalent diameters derived from 2D projections of highly irregular, droplet-like particles and compacted aggregates. A major consequence of this is that Dp/Dc becomes strongly sensitive to particle morphology, orientation on the substrate, and, potentially, how the particle dried. Thus, the authors are encouraged to consider using the mass (or volume) ratio of coating to core, which is particle morphology agnostic. As a added benefit, the EMBS–DDSCAT framework used by the authors provides explicit core and matrix volumes for the modeled particles, thereby making it straightforward to examine optical enhancement versus coating:core mass (or volume) ratio instead of versus Dp/Dc. Finally, as cited above, error analysis is strongly suggested for this metric.

**Reply: Thanks. In this study, to obtain the 3D diameter of particles, we established a correlation between the 2D diameter and the 3D diameter of particles using atomic force microscopy (AFM). This correlation enables the**

estimation of the 3D diameter of soot-containing particles. The employment of the 3D diameter can reduce the influence of particle morphology and the substrate on the $D_p/D_c$ ratio. To make the results more straightforward, we calculated the volume ratio of soot coatings to soot cores ($V_{coating}/V_c$) in Table S2. The error (i.e., standard deviation) of $D_p/D_c$ was also provided in Figure 7.

P8 L200-201: "The volume ratio of soot coatings to soot cores ($V_{coating}/V_c$) was further calculated according to the $D_p/D_c$."

P16 L417-419: "The $D_p/D_c$ and $V_{coating}/V_c$ ratios of transboundary soot-containing particles were calculated to reflect the coating thickness of soot particles and to quantify the aging degree of soot particles (Figure 7 and Table S2)."

P16 L422: "Correspondingly, the mean $V_{coating}/V_c$ ratios remained at 12-13 and 22-24 (Table S2)."

P16 L426-427: "Their mean $V_{coating}/V_c$ ratios also increased from 12 and 22 in the NCP to 21 and 39 in the YRD (Table S2)."

P17 L449-451: "Consistently, the mean $V_{coating}/V_c$ ratios increased from 13 for the partly-coated structure and 24 for the embedded structure to 18 and 83 (Table S2)."

**Table S2.** The mean volume ratio of soot coatings to soot cores ($V_{coating}/V_c$) in two types of transboundary transport models from the NCP to the YRD.

| Transport pathway | Region | Particle type | $V_{coating}/V_c$ |
|---|---|---|---|
| Inland | NCP | Partly-coated | 12 |
| | | Embedded | 22 |
| | YRD | Partly-coated | 21 |
| | | Embedded | 39 |
| Sea | NCP | Partly-coated | 13 |
| | | Embedded | 24 |
| | YRD | Partly-coated | 18 |
| | | Embedded | 83 |

**5.** There is the implicit assumption that the coatings are non-absorbing (1.53+0i). While this might be appropriate or sulfate/nitrate rich shells it may not be nearly as robust if organic coatings contain brown carbon. Given that the authors are focusing on relative effects of multicore geometry vs. coating thickness, this may be an acceptable assumption, but they are encouraged to explicitly cite this simplifying assumption. Further, a sensitivity analysis as function of center vs. periphery vs. random position is encouraged to support the robustness of a 72% reduction claim.

**Reply: Yes, if coatings on soot contain brown carbon, the optical characteristics of both coatings and soot-containing particles may change. As the reviewer commented, this study mainly focuses on relative effects of multicore geometry vs. coating thickness, so we simply assume the coating is non-absorbing. To clarify it, we have emphasized this simplified assumption for the optical simulation. We also added the sensitivity test to show the credibility of 72% $\Delta E_{abs}/\Delta(D_p/D_c)$ reduction (Figure S9).**

P20 L530-532: "Moreover, the coatings of soot cores were assumed to be non-absorbing materials in the optical calculation."

**P23 L634-636: "Based on the optical simulation (assuming that coatings on soot are non-absorbing), transboundary soot-containing particles transported through the inland pathway exhibited a $\Delta E_{abs}/\Delta(D_p/D_c)$ of 0.6."**

**P21-22 L575-583: "A sensitivity test was conducted for the $\Delta E_{abs}/\Delta(D_p/D_c)$ reduction caused by the transport pathway change through varying the embedded soot core position in host particles, as shown in Figure S9. It was found that the $\Delta E_{abs}/\Delta(D_p/D_c)$ reduction is 68% when soot cores are randomly distributed in host particles (Figure S9). This is close to the 72% reduction calculated when soot cores are distributed at the periphery of host particles (Figure S9). Because over 80% of the embedded soot cores were observed to be distributed at the periphery of transboundary particles, and the remainder was primarily randomly distributed, the 72% $\Delta E_{abs}/\Delta(D_p/D_c)$ reduction can be considered reliable."**

[Figure]

**Figure S9.** The $\Delta E_{abs}/\Delta(D_p/D_c)$ reduction of transboundary soot-containing particles with different core positions following the transport pathway change from the inland to the sea.

6. One of the take-home messages of this paper is that $\Delta E_{abs}/\Delta(D_p/D_c)$ is reduced by ~72% for sea-path soot because cloud processing generates multicore particles. This result is a model-derived result. Thus it might be useful to emphasize that $\Delta E_{abs}/\Delta(D_p/D_c)$ is a conceptual metric that depends on the chosen definition of Dp and Dc (e.g., equivalent sphere vs. outer droplet edge, etc.). This would underscore that while model–model comparison is likely

robust, model–measurement comparison would likely require careful matching of definitions.

**Reply: We emphasized the $\Delta E_{abs}/\Delta(D_p/D_c)$ as a conceptual metric and its applicability.**

**P22 L590-593: "It should be noted that the $\Delta E_{abs}/\Delta(D_p/D_c)$ derived from optical simulation is a conceptual metric, which depends on the chosen definition of $D_p$ and $D_c$ (e.g., ESD). While model–model comparisons are likely robust, model–measurement comparisons would likely require careful matching of definitions."**

7.  This reviewer was surprised by the rapid growth for the inland-route particles reported to reach 0.5–0.7 µm within a single transport episode. This is much larger than typically observed for non-cloud, early-age haze. To what degree is this a reflection of measurement bias - either through choice of methodology and/or selected size range vs actual growth that reflects intense secondary production? This is not to say that measurement bias invalidates the inland–sea comparison, but they suggest that the absolute size and coating metrics should be interpreted cautiously. The authors are encouraged to talk about this, especially in their comparisons with previous results that focused on sub-micron particles.

    **Reply: We apologize for any confusion caused to the reviewer regarding the interpretation that soot-containing particles transported through the inland pathway were mainly distributed within the broad size range of 500–700 nm in the YRD. Figure 5 actually shows the number percentage of soot-containing particles across different size bins. While the percentages were indeed high in both the 500–600 nm and 600–700 nm bins, the absolute number of soot-containing particles in the 500–600 nm range was approximately twice higher than that in the 600–700 nm range. This indicates that soot-containing particles were predominantly distributed around 500–600 nm, though this result remained a rough range. Further, we analyzed the number size distribution of soot-containing particles (Figure 6).**

Based on the size distribution, the dominant size of soot-containing particles can be accurately determined. We added some discussion and revised the result as follows.

**P15-16 L402-411: "The peak diameter at 505 nm for embedded soot-containing particles transported through the inland pathway is close to ~550 nm of aged soot-containing particles during regional haze reported by Wang et al. (2019). Although number fractions of these embedded soot-containing particles with 1 core were high in both the 500-600 nm and 600-700 nm bins (Figure 5b), the absolute number in the 500-600 nm range was approximately twice higher than that in the 600-700 nm range. As a result, the preponderant soot-containing particles in the YRD, i.e., embedded ones with 1 core (inland) and ≥ 3 cores (sea), dominated in the coarser size range of 500-600 nm and in the much coarser size range of > 1600 nm, respectively (Figure 5b, d)."**

[Figure]

**Figure 5.** Number fractions of soot-containing particles with different mixing structures and numbers of soot cores in different size bins in two types of transboundary transport models from the NCP to the YRD. (a-b) Soot-containing particles transported through the inland pathway. (c-d) Soot-containing particles transported through the sea pathway.

8. Finally, this reviewer would also like urge caution on any quantitative comparisons to the submicron BC literature. It is well-documented that sub-micron soot acquires nm-scale coatings through condensation and heterogeneous chemistry with concomitant restructuring processes, whereas the super-micron particles, the size regime that this paper is focused on, form through cloud activation, aqueous growth, and droplet-scale coalescence, requiring a very different aging pathway. As a consequence, caution needs to be exercised when comparing metrics like Dp/Dc, Df, and absorption enhancement from one regime to the other. Hence, the authors are encouraged to be explicit in treating their comparisons with sub-micron studies as qualitative.

**Reply: We have carefully considered the comments and revised some quantitative comparisons into qualitative ones.**

**P17 L453-455: "Moreover, Xu et al. (2020) showed a relatively high $D_p/D_c$ increase proportion of soot-containing particles during the transportation of dust storms from China across the East China Sea to Japan."**

**P20 L548-551: "In addition, comparable radiative absorption changes for soot-containing particles with different numbers of soot cores were observed during the transformation of soot core positions (Zhang et al., 2022)."**

9. Page 6, line 155; "…sampling duration of individual particles needs to be adjusted from 30 s to 15 min according to current PM2.5 concentrations." Please cite the concentration range that dictated sampling durations that spanned the 30-sec to 15-min range.

**Reply: We added the concentration range as follows.**

**P6 L155-157: "To avoid particles overlapping on the substrate, the sampling duration of individual particles needs to be adjusted from 30 s to 15 min according to current PM$_{2.5}$ concentrations from 17 μg m$^{-3}$ to 320 μg m$^{-3}$."**

10. Page 15 (line 406)/Page 16 (line 407-408): "Soot particles have been demonstrated to promote the formation of secondary aerosols around them via heterogeneous or aqueous-phase reactions". Soot is not a catalyst as implied in this sentence. Soot is chemically inert. Their primary role is as a non-reactive, insoluble substrate upon which material can be condensed onto. Please reword.

**Reply: We reworded this sentence as follows.**

**P16-17 L435-437: "Soot particles have been demonstrated to provide a substrate for the formation of secondary aerosols via heterogeneous or aqueous-phase reactions (Farley et al., 2023; Han et al., 2013; Zhu et al., 2025)."**

11. Figure 7: Please make the scale on the ordinate the same. It will help underscore just how difference the two ratios are. Also, as stated above, the authors might be serious consideration for using the mass ratio instead of Dp/Dc due to its independence of particle morphology.

**Reply: The scales on the ordinates in Figure 7 were made same. We also added the V$_{coating}$/V$_c$ ratio in Table S2.**

**References**

[revised manuscript text omitted]